# Towards Robust and Reliable Algorithmic Recourse

**Sohini Upadhyay**[*]
Harvard University
supadhyay@g.harvard.edu

**Shalmali Joshi**[*]
Harvard University
shalmali@seas.harvard.edu

**Himabindu Lakkaraju**
Harvard University
hlakkaraju@hbs.harvard.edu

## Abstract

As predictive models are increasingly being deployed in high-stakes decision making (e.g., loan approvals), there has been growing interest in post-hoc techniques which provide recourse to affected individuals. These techniques generate recourses under the assumption that the underlying predictive model does not change. However, in practice, models are often regularly updated for a variety of reasons (e.g., dataset shifts), thereby rendering previously prescribed recourses ineffective. To address this problem, we propose a novel framework, RObust Algorithmic Recourse (ROAR), that leverages adversarial training for finding recourses that are robust to model shifts. To the best of our knowledge, this work proposes the first ever solution to this critical problem. We also carry out theoretical analysis which underscores the importance of constructing recourses that are robust to model shifts: 1) We quantify the probability of invalidation for recourses generated without accounting for model shifts. 2) We prove that the additional cost incurred due to the robust recourses output by our framework is bounded. Experimental evaluation on multiple synthetic and real-world datasets demonstrates the efficacy of the proposed framework.

## 1 Introduction

Over the past decade, machine learning (ML) models are increasingly being deployed to make a variety of highly consequential decisions ranging from bail and hiring decisions to loan approvals. Consequently, there is growing emphasis on designing tools and techniques which can provide *recourse* to individuals who have been adversely impacted by predicted outcomes [30]. For example, when an individual is denied a loan by a predictive model deployed by a bank, they should be provided with reasons for this decision, and also informed about what can be done to reverse it. When providing a recourse to an affected individual, it is absolutely critical to ensure that the corresponding decision making entity (e.g., bank) is able to honor that recourse and approve any re-application that fully implements the recommendations outlined in the prescribed recourse Wachter et al. [31].

Several approaches in recent literature tackled the problem of providing recourses by generating *local* (instance level) counterfactual explanations [2] [31, 26, 12, 21, 18]. For instance, Wachter et al. [31] proposed a gradient based approach which finds the closest modification (counterfactual) that can result in the desired prediction. Ustun et al. [26] proposed an efficient integer programming based approach to obtain *actionable* recourses in the context of linear classifiers. There has also been some recent research that sheds light on the spuriousness of the recourses generated by counterfactual/contrastive explanation techniques [31, 26] and advocates for causal approaches [3, 14, 15].

All the aforementioned approaches generate recourses under the assumption that the underlying predictive models do not change. This assumption, however, may not hold in practice. Real world

---

[*]Equal contribution

[2]Note that counterfactual explanations [31], contrastive explanations [13], and recourse [26] are used interchangeably in prior literature. Counterfactual/contrastive explanations serve as a means to provide recourse to individuals with unfavorable algorithmic decisions. We use these terms interchangeably to refer to the notion introduced and defined by Wachter et al. [31]

settings are typically rife with different kinds of distribution shifts (e.g, temporal shifts) [22]. In order to ensure that the deployed models are accurate despite such shifts, these models are periodically retrained and updated. Such model updates, however, pose severe challenges to the validity of recourses because previously prescribed recourses (generated by existing algorithms) may no longer be valid once the model is updated. Recent work by Rawal et al. [24] has, in fact, demonstrated empirically that recourses generated by state-of-the-algorithms are readily invalidated in the face of model shifts resulting from different kinds of dataset shifts (e.g., temporal, geospatial, and data correction shifts). Their work underscores the importance of generating recourses that are robust to changes in models i.e., model shifts, particularly those resulting from dataset shifts. However, none of the existing approaches address this problem.

In this work, we propose a novel algorithmic framework, RObust Algorithmic Recourse (ROAR) for generating instance level recourses (counterfactual explanations) that are robust to changes in the underlying predictive model. To the best of our knowledge, this work makes the first attempt at generating recourses that are robust to model shifts. To this end, we propose a novel minimax objective that can be used to construct robust actionable recourses while minimizing the recourse costs. Second, we propose a set of model shifts that captures our intuition about the kinds of changes in the models to which recourses should be robust. Next, we outline an algorithm inspired by adversarial training to optimize the proposed objective. We also carry out theoretical analysis to establish the following results: i) we quantify the probability of invalidation for recourses generated without accounting for model shifts, and ii) we derive an upper bound on the relative increase in the costs incurred due to robust recourses (proposed by our framework) to the costs incurred by recourses generated from existing algorithms. Our theoretical results further establish the need for approaches like ours that generate actionable recourses that are robust to model shifts.

We evaluated our approach ROAR on real world data from financial lending and education domains, focusing on model shifts induced by the following kinds of distribution shifts – data correction shift, temporal shift, and geospatial shift. We also experimented with synthetic data to analyze how the degree of data distribution shifts and consequent model shifts affect the robustness and validity of the recourses output by our framework as well as the baselines. Our results demonstrate that the recourses constructed using our framework, ROAR, are substantially more robust ($67 - 100\%$) to changes in the underlying predictive models compared to those generated using state-of-the-art recourse finding technqiues. We also find that our framework achieves such a high degree of robustness without sacrificing the validity of the recourses w.r.t. the original predictive model or substantially increasing the costs associated with realizing the recourses.

## 2   Related Work

Our work lies at the intersection of algorithmic recourse and adversarial robustness. Below, we discuss related work pertaining to each of these topics.

**Algorithmic recourse**   As discussed in Section 1, several approaches have been proposed to construct algorithmic recourse for predictive models [31, 26, 12, 21, 18, 3, 14, 15, 7]. These approaches can be broadly characterized along the following dimensions [29]: the level of access they require to the underlying predictive model (black box vs. gradients), if and how they enforce sparsity (only a small number of features should be changed) in counterfactuals, if counterfactuals are required to lie on the data manifold or not, if underlying causal relationships should be accounted for when generating counterfactuals or not, whether the output should be multiple diverse counterfactuals or just a single counterfactual. While the aforementioned approaches have focused on generating instance level counterfactuals, there has also been some recent work on generating global summaries of model recourses which can be leveraged to audit ML methods [23]. More recently, Rawal et al. [24] demonstrated that recourses generated by state-of-the-art algorithms are readily invalidated due to model shifts resulting from different kinds of dataset shifts. They argued that model updation is very common place in the real world, and it is important to ensure that recourses provided to affected individuals are robust to such updates. Similar arguments have been echoed in several other recent works [28, 13, 20]. While there has been some recent work that explores the construction of other kinds of explanations (feature attribution and rule based explanations) that are robust to dataset shifts [16], our work makes the first attempt at tackling the problem of constructing recourses that are robust to model shifts.

**Adversarial Robustness** The techniques that we leverage in this work are inspired by the adversarial robustness literature. Wachter et al. were the first to remark on similarities between counterfactual generation and adversarial attacks, but did not leverage this connection to develop robust recourse [31]. It is now well established that ML models are vulnerable to adversarial attacks [10, 4, 2]. The adversarial training procedure was recently proposed as a defense against such attacks [19, 1, 32]. This procedure optimizes a minimax objective that captures the worst-case loss over a given set of perturbations to the input data. At a high level, it is based on gradient descent; at each gradient step, it solves an optimization problem to find the worst-case perturbation, and then computes the gradient at this perturbation. In contrast, our training procedure optimizes a minimax objective that captures the worst-case over a given set of model perturbations (thereby simulating model shift) and generates recourses that are valid under the corresponding model shifts. This training procedure is novel and possibly of independent interest.

## 3 Our Framework: RObust Algorithmic Recourse

In this section, we detail our framework, RObust Algorithmic Recourse (ROAR). First, we introduce some notation and discuss preliminary details about the algorithmic recourse problem setting. We then introduce our objective function, and discuss how to operationalize and optimize it efficiently.

### 3.1 Preliminaries

Let us assume we are given a predictive model $\mathcal{M} : \mathcal{X} \to \mathcal{Y}$, where $\mathcal{X} \subseteq \mathbb{R}^d$ is the feature space, and $\mathcal{Y}$ is the space of outcomes. Let $\mathcal{Y} = \{0, 1\}$ where 0 and 1 denote an unfavorable outcome (e.g., loan denied) and a favorable outcome (e.g., loan approved) respectively. Let $x \in \mathcal{X}$ be an instance which received a negative outcome i.e., $\mathcal{M}(x) = 0$. The goal here is to find a recourse for this instance $x$ i.e., to determine a set of changes $\epsilon$ that can be made to $x$ in order to reverse the negative outcome. The problem of finding a recourse for $x$ involves finding a counterfactual $x' = x + \epsilon$ for which the black box outputs a positive outcome i.e., $\mathcal{M}(x') = \mathcal{M}(x + \epsilon) = 1$.

There are, however, a few important considerations when finding the counterfactual $x' = x + \epsilon$. First, it is desirable to minimize the cost (or effort) required to change $x$ to $x'$. To formalize this, let us consider a cost function $c : \mathcal{X} \times \mathcal{X} \to \mathbb{R}_+$. $c(x, x')$ denotes the cost (or effort) incurred in changing an instance $x$ to $x'$. In practice, some of the commonly used cost functions are $\ell_1$ or $\ell_2$ distance [31], log-percentile shift [26], and costs learned from pairwise feature comparisons input by end users [23]. Furthermore, since recommendations to change features such as gender or race would be unactionable, it is important to restrict the search for counterfactuals in such a way that only actionable changes are allowed. Let $\mathcal{A}$ denote the set of plausible or actionable counterfactuals.

Putting it all together, the problem of finding a recourse for instance $x$ for which $\mathcal{M}(x) = 0$ can be formalized as:

$$x' = \underset{x' \in \mathcal{A}}{\arg\min} \, c(x, x') \quad \text{s.t} \quad \mathcal{M}(x') = 1 \tag{1}$$

Eqn. 1 captures the generic formulation leveraged by several of the state-of-the-art recourse finding algorithms. Typically, most approaches optimize the unconstrained and differentiable relaxation of Eqn. 1 which is given below:

$$x' = \underset{x' \in \mathcal{A}}{\arg\min} \, \ell(\mathcal{M}(x'), 1) + \lambda \, c(x, x') \tag{2}$$

where $\ell : \mathcal{Y} \times \mathcal{Y} \to \mathbb{R}_+$ denotes a differentiable loss function (e.g., binary cross entropy) which ensures that gap between $\mathcal{M}(x')$ and favorable outcome 1 is minimized, and $\lambda > 0$ is a trade-off parameter.

### 3.2 Formulating Our Objective

As can be seen from Eqn. 2, state-of-the-art recourse finding algorithms rely heavily on the assumption that the underlying predictive model $\mathcal{M}$ does not change. However, predictive models deployed in the real world often get updated. This implies that individuals who have acted upon previously prescribed recourses are no longer guaranteed a favorable outcome once the model is updated. To address this critical challenge, we propose a novel minimax objective function which generates counterfactuals that minimize the worst-case loss over plausible model shifts. We arrived at this approach after considering the following alternatives: (a) Update the predictive model as desired but ensure that individuals who were previously prescribed recourse will still be guaranteed a favorable outcome. (b)

Update the predictive model while including constraints to ensure that previously offered recourses are still valid. Note that both of these scenarios would potentially incur huge monetary losses to relevant stakeholders (e.g, banks), hurting the adoption of these approaches. In case (a), banks may be required to guarantee credit to customers that are potentially not creditworthy under the new model and thereby risk losing money. In case (b), access to model training is assumed. Furthermore, training a predictive model under these constraints may be suboptimal and not reflective of the current data distribution, thereby accruing larger errors under the shifted population. There are no incentives for stakeholders such as banks to adopt such practices which could potentially lead to huge monetary losses. While the optimal approach may vary on a case by case basis, we propose our method to avoid the aforementioned pitfalls outlined in cases (a) and (b).

To formalize our proposed approach, let $\Delta$ denote the set of plausible model shifts and let $\mathcal{M}_\delta$ denote a shifted model where $\delta \in \Delta$. Our objective function for generating robust recourse $x''$ for a given instance $x$ can be written as:

$$x'' = \underset{x'' \in \mathcal{A}}{\arg\min} \max_{\delta \in \Delta} \ell(\mathcal{M}_\delta(x''), 1) + \lambda c(x, x'') \tag{3}$$

where cost function $c$ and loss function $l$ are as defined in Section 3.1.

**Choice of $\Delta$**   Predictive models deployed in the real world are often updated regularly to handle data distribution shifts [22]. Since these models are updated regularly, it is likely that they undergo small (and not drastic) shifts each time they are updated. To capture this intuition, we consider the following two choices for the set of plausible model shifts $\Delta$:

$\Delta = \{\delta \in \mathbb{R}^n \mid \delta_{min} \leq \delta_i \leq \delta_{max} \forall i \in \{1 \cdots n\}\}$.

$\Delta = \{\delta \in \mathbb{R}^n \mid \|\delta\|_p \leq \delta_{max}\}$

where $p \geq 1$. Note that perturbations $\delta \in \Delta$ can be considered as operations either on the parameter space or on the gradient space of $\mathcal{M}$. While the first choice of $\Delta$ presented above allows us to restrict model shifts within a small range, the second choice allows us to restrict model shifts within a norm-ball. Alternate formulations of the first include incorporating domain knowledge to set $\delta_{min}$ and $\delta_{max}$ per feature. These kinds of shifts can effectively capture small changes to both parameters (e.g., weights of linear models) as well as gradients. Next, we describe how to optimize the objective in Eqn. 3 and construct robust recourses.

### 3.3   Optimizing Our Objective

While our objective function, the choice of $\Delta$, and the perturbations $\delta \in \Delta$ we introduce in Section 3.2 are generic enough to handle shifts to both parameter space as well as the gradient space of any class of predictive models $\mathcal{M}$, we solve our objective for a linear approximation $f$ of $\mathcal{M}$. The procedure that we outline here remains generalizable even for non-linear models because local behavior of a given non-linear model can be approximated well by fitting a local linear model [25]. Note that such approximations have already been explored by existing algorithmic recourse methods [26, 23]. Let the linear approximation, which we denote by $f$ be parameterized by $w \in \mathcal{W}$. We make this parametrization explicit by using a subscript notation: $f_w$. We consider model shifts represented by perturbations to the model parameters $w \in \mathcal{W}$. In the case of linear models, these can be operationalized as additive perturbations $\delta \in \Delta$ to $w$. We will represent the resulting shifted classifier by $f_{w+\delta}$. Our objective function (Eqn. 3) can now be written in terms of this linear approximation $f$ as:

$$x'' = \underset{x'' \in \mathcal{A}}{\arg\min} \max_{\delta \in \Delta} \ell(f_{w+\delta}(x''), 1) + \lambda c(x, x'') \tag{4}$$

Notice that the objective function defined in Equation 4 is similar to that of adversarial training [19]. However, in our framework, the perturbations are applied to model parameters as opposed to data samples. These parallels help motivate the optimization procedure for constructing recourses that are robust to model shifts. We outline the optimization procedure that we leverage to optimize our minimax objective (Eqn. 4) in Algorithm 1.

Algorithm 1 proceeds in an iterative manner where we first find a perturbation $\hat{\delta} \in \Delta$ that maximizes the chance of invalidating the current estimate of the recourse $x''$, and then we take appropriate gradient steps on $x''$ to generate a valid recourse. This procedure is executed iteratively until the objective function value (Eqn. 4) converges.

---

**Algorithm 1** Our Optimization Procedure

---

**Input:** $x$ s.t. $f_w(x) = 0, f_w, \lambda > 0, \Delta$, learning rate $\alpha > 0$.
**Initialize** $x'' = x, g = 0$
**repeat**
    $\hat{\delta} = \arg\max_{\delta \in \Delta} \ell(f_{w+\delta}(x''), 1)$
    $g = \nabla \Big[ \ell(f_{w+\hat{\delta}}(x''), 1) + \lambda c(x'', x) \Big]$
    $x'' \mathrel{-}= \alpha g$
**until** convergence
Return $x''$

---

## 4 Theoretical Analysis

Here we carry out theoretical analysis to shed light on the benefits of our framework ROAR. More specifically: 1) We quantify the probability that recourses generated without accounting for model shifts are likely to be invalidated. 2) We prove that the additional cost incurred due to the robust recourses output by our framework is bounded.

We first characterize how recourses that do not account for model shifts (i.e., recourses output by state-of-the-art algorithms) fare when true model shifts can be characterized as additive shifts to model parameters. Specifically, we quantify the likelihood that recourses generated without accounting for model shifts will be invalidated (even if they lie on the original data manifold), under certain conditions.

**Theorem 1.** *For a given instance $x \sim \mathcal{N}(\mu, \Sigma)$, let $x'$ be the recourse that lies on the original data manifold (conditioned on the event that $\mathcal{M}(x') > 0.5$) and is obtained without accounting for model shifts. Let $\Sigma = UDU^T$. Then, for some* true *model shift $\delta$, such that, $\frac{w^T \mu}{(w+\delta)^T \mu} \geq \frac{\|\sqrt{D}Uw\|}{\|\sqrt{D}U(w+\delta)\|}$, and*
$\sqrt{\frac{2e}{\pi}} \frac{\sqrt{\beta-1}}{\beta} \exp^{-\beta \frac{(w^T\mu)^2}{4\|\sqrt{D}Uw\|^2}} \geq erfc(-\frac{(w+\delta)^T\mu}{\sqrt{2}\|w+\delta\|})$, *for $\beta \geq 1$, the probability that $x'$ is invalidated on $f_{w+\delta}$ is at least: $\frac{1}{2}\sqrt{\frac{2e}{\pi}} \frac{\sqrt{\beta-1}}{\beta} \exp^{-\beta \frac{(w^T\mu)^2}{4\|\sqrt{D}Uw\|^2}} - \frac{1}{2}erfc(-\frac{(w+\delta)^T\mu}{\sqrt{2}\|w+\delta\|})$ where erfc is the complementary gaussian error function.*

*Proof Sketch.* Under the assumption that $x' \sim \mathcal{N}(\mu, \Sigma)$, a recourse is invalid under a model shift if it is valid under the original model and invalid under the shifted model. This allows us to define the region where $x'$ can be invalidated:

$$\Omega = \{x' : w^T x' > 0 \cap (w+\delta)^T x' \leq 0\}$$

The probability that $x'$ is invalidated can be obtained by integrating over $\Omega$ under the PDF of $\mathcal{N}(\mu, \Sigma)$.

We can then transform $x'$ and correspondingly $\Omega$, to simplify this integration over a 1-dimensional Gaussian random variable. That is,

$$P(x \text{ is invalidated}) = \frac{1}{\sqrt{(2\pi)}} \int_{c_1}^{c_2} \exp\left(-\frac{1}{2}s^2\right) ds \tag{5}$$

where $\Omega_s = \{s : [c_1, c_2]\}$, $c_1 = \frac{-w^T\mu}{\|\sqrt{D}Uw\|}$ and $c_2 = \frac{-(w+\delta)^T\mu}{\|\sqrt{D}U(w+\delta)\|}$

The above quantity can be represented as a difference in the Gaussian error function allowing us to exactly quantify the invalidation probability under our assumptions. Using the lower bounds on the complementary gaussian error function [9] from Chang et al. [5], we obtain our lower bound. To derive the lower bound, we add an extra condition that $\sqrt{\frac{2e}{\pi}} \frac{\sqrt{\beta-1}}{\beta} \exp^{-\beta \frac{(w^T\mu)^2}{4\|\sqrt{D}Uw\|^2}} \geq erfc(-\frac{(w+\delta)^T\mu}{\sqrt{2}\|w+\delta\|})$, mainly to confirm that the lower bound on the first term still dominates the second term. Both conditions restricts the types of shift for which the bound can be derived. Note that $\beta$ can be optimized away to improve the lower bound. Detailed proof is provided in the Appendix. Discussion about other distributions (e.g., Bernoulli, Uniform, Categorical) is included in the Appendix. □

Next we characterize how much more costly recourses can be when they are trained to be robust to model perturbations or model shifts. In the following theorem, we show that the cost of robust recourses is bounded relative to the cost of recourses that do not account for model shifts.

**Theorem 2.** *Let $x \in \mathcal{X}$, and $x \sim \nu$ where $\nu$ is a distribution such that $\mathbb{E}_\nu[x] = \mu < \infty$, where $\mathcal{X}$ is a metric space $(\mathcal{X}, d(\cdot, \cdot))$ and $d : \mathcal{X} \times \mathcal{X} \to \mathbb{R}_+$. Let $d \triangleq \ell_2$ and assume that $(\mathcal{X}, d)$ has bounded diameter $D = \sup_{x,x' \in \mathcal{X}} d(x, x')$. Let recourses obtained without accounting for model shifts and constrained to the manifold be denoted by $x' \sim \nu$, and robust recourses be denoted by $x''$. Let $\delta > 0$ be the shift that maximizes Eq. 3 for sample $x$ corresponding to $x''$. Further assume that the ROAR objective (Equation 3) is convex in $x''$ for a fixed $\delta$. For $\ell \triangleq \ell_{log}$ (the cross-entropy loss), some $0 < \eta' \ll 1$, w.h.p. $(1 - \eta')$, we have that:*

$$c(x'', x) - c(x', x) \le \frac{1}{\lambda} \frac{1}{\|w + \delta\|} \mathbb{E}_\nu[\exp -\phi(w + \delta)^T x'] + \sqrt{\frac{D^2}{2} \log\left(\frac{1}{\eta'}\right)}\right) \tag{6}$$

*Proof Sketch.* By definition, any recourse $x'$ generated without accounting for model shifts will have a higher loss for Equation 3 compared to the robust recourse $x''$ (note that finding the global minimizer is not guaranteed by Algorithm 1).

Using this insight, and convexity in $x'$ for fixed $\delta$, we can bound the cost difference between the robust and non-robust recourse by a 1-Lipschitz function (i.e. the logistic function):

$$c(x'', x) - c(x', x) \le \frac{1}{\lambda \|w + \delta\|} \log\{1 + \exp -(w + \delta)^T x'\}$$

Assuming a bounded metric on $\mathcal{X}$, we can upper bound the RHS using Lemma 2 from van Handel [27] which gives us our bound. Detailed proof including special cases when $\nu$ is Gaussian, is provided in the Appendix. □

This result suggests that the additional cost of recourse is bounded by the amount of shift admissible in Equation 3. Note that Theorem 2 applies for general distributions so long as the mean is finite, which is the case for most commonplace distributions like Gaussian, Bernoulli, Multinomial etc. While Theorem 1 demonstrates the probability that a recourse will be invalidated for Gaussian distributions, we refer the reader to the Appendix B.1 for a discussion of other distributions, e.g. Bernoulli, Uniform, Categorical.

## 5   Experiments

Here we discuss the detailed experimental evaluation of our framework, ROAR. First, we evaluate how robust the recourses generated by our framework are to model shifts caused by real world data distribution shifts. We also assess the validity of the recourses generated by our framework w.r.t. the original model, and further analyze the average cost of these recourses. Next, using synthetic data, we analyze how varying the degree (magnitude) of data distribution shift impacts the robustness and validity of the recourses output by our framework and other baselines.

### 5.1   Experimental Setup

**Real world data**   We evaluate our framework on model shifts induced by real world data distribution shifts. To this end, we leverage three real world datasets which capture different kinds of data distribution shifts, namely, temporal shift, geospatial shift, and data correction shift [24]. Our first dataset is the widely used and publicly available **German credit** dataset [8] from the UCI repository. This dataset captures demographic (age, gender), personal (marital status), and financial (income, credit duration) details of about 1000 loan applicants. Each applicant is labeled as either a good customer or a bad customer depending on their credit risk. Two versions of this dataset have been released, with the second version incorporating corrections to coding errors in the first dataset [11]. Accordingly, this dataset captures the **data correction** shift. Our second dataset is the **Small Business Administration (SBA) case** dataset [17]. This dataset contains information pertaining to 2102 small business loans approved by the state of California during the years of $1989 - 2012$, and captures **temporal shifts** in the data. It comprises of about 24 features capturing various details of the small businesses including zip codes, business category (real estate vs. rental vs. leasing), number of jobs created, and financial status of the business. It also contains information about whether a business

has defaulted on a loan or not which we consider as the class label. Our last dataset contains **student performance** records of $649$ students from two Portuguese secondary schools, Gabriel Pereira (GP) and Mousinho da Silveira (MS) [8, 6], and captures **geospatial shift**. It comprises of information about the academic background (grades, absences, access to internet, failures etc.) of each student along with other demographic attributes (age, gender). Each student is assigned a class label of above average or not depending on their final grade.

**Synthetic data** We generate a synthetic dataset with $1K$ samples and two dimensions to analyze how the degree (magnitude) of data distribution shifts impacts the robustness and validity of the recourses output by our framework and other baselines. Each instance $x$ is generated as follows: First, we randomly sample the class label $y \in \{0, 1\}$ corresponding to the instance $x$. Conditioned upon the value of $y$, we then sample the instance $x$ as: $x \sim \mathcal{N}(\mu_y, \Sigma_y)$. We choose $\mu_0 = [-2, -2]^T$ and $\mu_1 = [+2, +2]^T$, and $\Sigma_0 = \Sigma_1 = 0.5\mathbf{I}$ where $\mu_0$, $\Sigma_0$ and $\mu_1$, $\Sigma_1$ denote the means and covariance of the Gaussian distributions from which instances in class 0 and class 1 are sampled respectively. A scatter plot of the samples resulting from this generative process and the decision boundary of a logistic regression model fit to this data are shown in Figure 1a. In our experimental evaluation, we consider different kinds of shifts to this synthetic data:

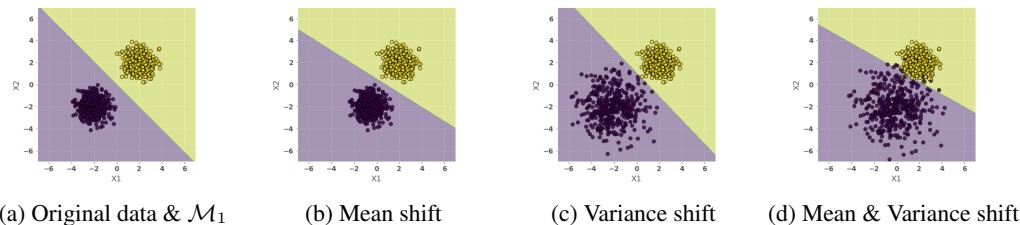

| (a) Original data & $\mathcal{M}_1$ | (b) Mean shift | (c) Variance shift | (d) Mean & Variance shift |

Figure 1: Synthetic data and examples of model shift. From left to right we have (a) original synthetic dataset, (b) shifted data and decision boundary after mean shift, (c) shifted data and decision boundary after variance shift, and (d) shifted data and decision boundary after mean and variance shift

**(i) Mean shift**: To generated shifted data, we leverage the same approach as above but shift the mean of the Gaussian distribution associated with class 0 i.e., $x \sim \mathcal{N}(\mu'_y, \Sigma_y)$ where $\mu'_0 = \mu_0 + [\alpha, 0]^T$ and $\mu'_1 = \mu_1$. Note that we only shift the mean of one of the features of class 0 so that the slope of the decision boundary of a linear model we fit to this shifted data changes (relative to the linear model fit on the original data), while the intercept remains the same. Figure 1b shows shifted data with $\alpha = 1.5$.

**(ii) Variance shift**: Here, we leverage the same generative process as above, but instead of shifting the mean, we shift the variance of the Gaussian distribution associated with class 0 i.e., i.e., $x \sim \mathcal{N}(\mu_y, \Sigma'_y)$ where $\Sigma'_0 = (1 + \beta)\Sigma_0$ and $\Sigma'_1 = \Sigma'_1$ for some increment $\beta \in \mathbb{R}$. The net result here is that the intercept of the decision boundary of a linear model we fit to this shifted data changes (relative to the linear model fit on the original data), while the slope remains unchanged. Figure 1c shows shifted data with $\beta = 3$.

**(ii) Mean and variance shift**: Here, we change both the mean and variance of the Gaussian distribution associated with class 0 simultaneously (Figure 1d). It can be seen that there are noticeable changes to both the slope and intercept of the decision boundary compared to Figure 1a.

**Predictive models** We generate recourses for a variety of linear and non-linear models: deep neural networks (DNNs), SVMs, and logistic regression (LR). Here, we present results for a 3-layer DNN and LR; remaining results are included in the Appendix. Results presented here are representative of those for other model families.

**Baselines** We compare our framework, ROAR, to the following state-of-the-art baselines: (i) counterfactual explanations (CFE) framework outlined by Wachter et al. [31], (ii) actionable recourse (AR) in linear classification [26], and (iii) causal recourse framework (MINT) proposed by Karimi et al. [14]. While CFE leverages gradient computations to find counterfactuals, AR employs a mixed integer programming based approach to find counterfactuals that are actionable. The MINT framework operates on top of existing approaches for finding nearby counterfactuals. We use the MINT framework on top of CFE and ROAR and refer to these two approaches as MINT and ROAR-MINT respectively. As the MINT framework requires access to the underlying causal graph, we

experiment with MINT and ROAR-MINT only on the German credit dataset for which such a causal graph is available.

**Cost functions**  Our framework, ROAR, and all the other baselines we use rely on a cost function $c$ that measures the cost (or effort) required to act upon the prescribed recourse. Furthermore, our approach as well as several other baselines require the cost function to be differentiable. So, we consider two cost functions in our experimentation: $\ell_1$ distance between the original instance and the counterfactual, and a cost function learned from pairwise feature comparison inputs (PFC) [13, 26, 23]. PFC uses the Bradley-Terry model to map pairwise feature comparison inputs provided by end users to the cost required to act upon the prescribed recourse for any given instance $x$. For more details on this cost function, please refer to Rawal and Lakkaraju [23]. In our experiments, we follow the same procedure as Rawal and Lakkaraju [23] and simulate the pairwise feature comparison inputs.

**Setting and implementation details**  We partition each of our synthetic and real world datasets into two parts: initial data ($D_1$) and shifted data ($D_2$). In the case of real world datasets, $D_1$ and $D_2$ can be logically inferred from the data itself – e.g., in case of the German credit dataset, we consider the initial version of the dataset as $D_1$ and the corrected version of the dataset as $D_2$. In the case of synthetic datasets, we generate $D_1$ and $D_2$ as described earlier where $D_2$ is generated by shifting $D_1$ (See "Synthetic data" in Section 5.1).

We use 5-fold cross validation throughout our real world and synthetic experiments. On $D_1$, we use 4 folds to train predictive models and the remaining fold to generate and evaluate recourses. We repeat this process 5 times and report averaged values of our evaluation metrics. We leverage $D_2$ only to train the shifted models $\mathcal{M}_2$. More details about the data splits, model training, and performance of the predictive models are included in the Appendix.

We use binary cross entropy loss and the Adam optimizer to operationalize our framework, ROAR. Our framework, ROAR, has the following parameters: the set of acceptable perturbations $\Delta$ (defined in practice by $\delta_{max}$) and the tradeoff parameter $\lambda$. In our experiments on evaluating robustness to real world shifts, we choose $\delta_{max} = 0.1$ given that continuous features are scaled to zero mean and unit variance. Furthermore, in each setting, we choose the $\lambda$ that maximizes the recourse validity of $\mathcal{M}_1$ (more details in Section 5.1 "Metrics" and Appendix). In case of our synthetic experiments where we assess the impact of the degree (magnitude) of data distribution shift, features are not normalized, so we do a grid search for both $\delta_{max}$ and $\lambda$. First, we choose the largest $\delta_{max}$ that maximizes the recourse validity of $\mathcal{M}_1$ and then set $\lambda$ in a similar fashion (more details in Appendix). We set the parameters of the baselines using techniques discussed in the original works [31, 14, 26] and employ a similar grid search approach if unspecified.

Following the precedents set forth in [26] and [23], we adapt AR and ROAR to non-linear models by first generating local linear approximations of these models using LIME [25]. We refer to these variants as AR-LIME and ROAR-LIME respectively.

**Metrics.**  We consider two metrics in our evaluation: 1) **Avg Cost** is defined as the average cost incurred to act upon the prescribed recourses where the average is computed over all the instances for which a given algorithm provides recourse. Recall that we consider two notions of cost in our experiments – $\ell_1$ distance between the original instance and the counterfactual, costs learned from pairwise feature comparisons (PFC) (See "Cost Functions" in Section 5.1). 2) **Validity** is defined as the fraction of instances for which acting upon the prescribed recourse results in the desired prediction. Note that validity is computed w.r.t. a given model.

## 5.2  Robustness to real world shifts

Here, we evaluate the robustness of the recourses output by our framework, ROAR, as well as the baselines. A recourse finding algorithm can be considered robust if the recourses output by the algorithm remain valid even if the underlying model has changed. To evaluate this, we first leverage our approach and other baselines to find recourses of instances in our test sets w.r.t. the initial model $\mathcal{M}_1$. We then compute the *validity* of these recourses w.r.t. the shifted model $\mathcal{M}_2$ which has been trained on the shifted data. Let us refer to this as $\mathcal{M}_2$ *validity*. The higher the value of $\mathcal{M}_2$ *validity*, the more robust the recourse finding method. Table 1 shows the $\mathcal{M}_2$ *validity* metric computed for different algorithms across different real world datasets.

It can be seen that recourse methods that use our framework, ROAR and ROAR-MINT, achieve the highest $\mathcal{M}_2$ *validity* across all datasets. In fact, methods that use our framework do almost twice as

| Model | Cost | Recourse | Correction Shift | | | Temporal Shift | | | Geospatial Shift | | |
|---|---|---|---|---|---|---|---|---|---|---|---|
| | | | Avg Cost | $\mathcal{M}_1$ Validity | $\mathcal{M}_2$ Validity | Avg Cost | $\mathcal{M}_1$ Validity | $\mathcal{M}_2$ Validity | Avg Cost | $\mathcal{M}_1$ Validity | $\mathcal{M}_2$ Validity |
| LR | L1 | CFE | $1.02 \pm 0.18$ | $1.00 \pm 0.00$ | $0.54 \pm 0.27$ | $3.57 \pm 1.14$ | $1.00 \pm 0.00$ | $0.31 \pm 0.09$ | $8.37 \pm 0.73$ | $0.98 \pm 0.03$ | $0.29 \pm 0.09$ |
| | | AR | $0.85 \pm 0.14$ | $1.00 \pm 0.00$ | $0.53 \pm 0.21$ | $1.50 \pm 0.28$ | $1.00 \pm 0.00$ | $0.16 \pm 0.06$ | $5.29 \pm 0.28$ | $1.00 \pm 0.00$ | $0.43 \pm 0.14$ |
| | | ROAR | $3.13 \pm 0.32$ | $1.00 \pm 0.00$ | $0.94 \pm 0.08$ | $3.14 \pm 0.25$ | $0.99 \pm 0.01$ | $\mathbf{0.98 \pm 0.02}$ | $10.88 \pm 1.67$ | $1.00 \pm 0.00$ | $\mathbf{0.67 \pm 0.19}$ |
| | | MINT | $4.73 \pm 1.56$ | $1.00 \pm 0.00$ | $0.93 \pm 0.07$ | NA | NA | NA | NA | NA | NA |
| | | ROAR-MINT | $6.77 \pm 0.35$ | $1.00 \pm 0.00$ | $\mathbf{1.00 \pm 0.00}$ | NA | NA | NA | NA | NA | NA |
| | PFC | CFE | $0.03 \pm 0.02$ | $1.00 \pm 0.00$ | $0.56 \pm 0.33$ | $0.24 \pm 0.09$ | $1.00 \pm 0.00$ | $0.26 \pm 0.11$ | $0.34 \pm 0.04$ | $1.00 \pm 0.00$ | $0.18 \pm 0.10$ |
| | | AR | $0.09 \pm 0.02$ | $1.00 \pm 0.00$ | $0.54 \pm 0.27$ | $0.11 \pm 0.02$ | $1.00 \pm 0.00$ | $0.09 \pm 0.05$ | $0.32 \pm 0.03$ | $1.00 \pm 0.00$ | $0.24 \pm 0.11$ |
| | | ROAR | $0.36 \pm 0.08$ | $1.00 \pm 0.00$ | $\mathbf{1.00 \pm 0.00}$ | $0.44 \pm 0.12$ | $0.99 \pm 0.01$ | $\mathbf{0.98 \pm 0.01}$ | $1.20 \pm 0.10$ | $1.00 \pm 0.00$ | $\mathbf{0.91 \pm 0.07}$ |
| | | MINT | $1.00 \pm 1.15$ | $1.00 \pm 0.00$ | $0.95 \pm 0.08$ | NA | NA | NA | NA | NA | NA |
| | | ROAR-MINT | $1.23 \pm 0.05$ | $1.00 \pm 0.00$ | $1.00 \pm 0.00$ | NA | NA | NA | NA | NA | NA |
| NN | L1 | CFE | $0.55 \pm 0.10$ | $1.00 \pm 0.00$ | $0.47 \pm 0.06$ | $3.78 \pm 0.68$ | $1.00 \pm 0.00$ | $0.52 \pm 0.09$ | $10.09 \pm 0.71$ | $1.00 \pm 0.00$ | $0.48 \pm 0.09$ |
| | | AR-LIME | $0.38 \pm 0.15$ | $0.16 \pm 0.10$ | $0.31 \pm 0.06$ | $1.39 \pm 0.13$ | $0.59 \pm 0.11$ | $0.65 \pm 0.17$ | $9.02 \pm 1.57$ | $0.76 \pm 0.06$ | $0.83 \pm 0.10$ |
| | | ROAR-LIME | $1.83 \pm 0.19$ | $0.78 \pm 0.06$ | $0.72 \pm 0.10$ | $4.90 \pm 0.24$ | $0.98 \pm 0.02$ | $\mathbf{0.97 \pm 0.02}$ | $21.05 \pm 3.58$ | $1.00 \pm 0.00$ | $\mathbf{0.97 \pm 0.03}$ |
| | | MINT | $2.24 \pm 1.25$ | $0.81 \pm 0.02$ | $0.63 \pm 0.11$ | NA | NA | NA | NA | NA | NA |
| | | ROAR-MINT | $8.59 \pm 1.70$ | $0.90 \pm 0.03$ | $\mathbf{0.84 \pm 0.04}$ | NA | NA | NA | NA | NA | NA |
| | PFC | CFE | $0.06 \pm 0.02$ | $1.00 \pm 0.00$ | $0.51 \pm 0.12$ | $0.19 \pm 0.06$ | $1.00 \pm 0.00$ | $0.50 \pm 0.13$ | $0.48 \pm 0.06$ | $1.00 \pm 0.00$ | $0.30 \pm 0.14$ |
| | | AR-LIME | $0.06 \pm 0.03$ | $0.49 \pm 0.11$ | $0.56 \pm 0.15$ | $0.11 \pm 0.01$ | $0.54 \pm 0.08$ | $0.62 \pm 0.12$ | $0.78 \pm 0.15$ | $0.84 \pm 0.06$ | $0.82 \pm 0.11$ |
| | | ROAR-LIME | $0.64 \pm 0.08$ | $0.85 \pm 0.07$ | $\mathbf{0.82 \pm 0.05}$ | $0.37 \pm 0.07$ | $0.99 \pm 0.01$ | $\mathbf{0.99 \pm 0.0}$ | $1.66 \pm 0.21$ | $1.00 \pm 0.00$ | $\mathbf{0.97 \pm 0.04}$ |
| | | MINT | $0.60 \pm 0.16$ | $0.82 \pm 0.07$ | $0.64 \pm 0.15$ | NA | NA | NA | NA | NA | NA |
| | | ROAR-MINT | $0.60 \pm 0.07$ | $0.91 \pm 0.04$ | $0.81 \pm 0.04$ | NA | NA | NA | NA | NA | NA |

Table 1: Avg Cost, $\mathcal{M}_1$ (original) validity, and $\mathcal{M}_2$ (shifted model) validity of recourses across different real world datasets. Recourses that leverage our framework ROAR are more robust (higher $\mathcal{M}_2$ validity) compared to those generated by existing baselines.

good compared to other baselines on this metric, indicating that ROAR based recourse methods are quite robust. After ROAR, MINT is the next best performing baseline with respect $\mathcal{M}_2$ validity. This may be explained by the fact that MINT accounts for the underlying causal graphs when generating recourses.

We also assess if the robustness achieved by our framework is coming at a cost i.e., by sacrificing *validity* on the original model or by increasing *avg cost*. Table 1 shows the results for the same. It can be seen that ROAR based recourses achieve higher than 95% $\mathcal{M}_1$ *validity* in all but two settings. We compute the *avg cost* of the recourses output by all the algorithms on various datasets and find that ROAR typically has a higher *avg cost* (both under $\ell_1$ and PFC cost functions) compared to CFE and AR baselines. As demonstrated through additional experiments in the Appendix, these relatively higher costs are expected given our Theorem 2 upper bound on ROAR cost. However, overall, MINT and ROAR-MINT seem to exhibit the highest *avg costs* and are the worst performing algorithms according to this metric. Since non-causal recourse methods assume independent features, and do not have to adhere to the underlying causal structure when finding counterfactuals, they can generate relatively lower cost counterfactuals even if those counterfactuals may not correspond to realistic data instances. This is likely one of the key reasons why we observe higher average costs in the causal recourse methods.

### 5.3 Impact of the degree of data distribution shift on recourses

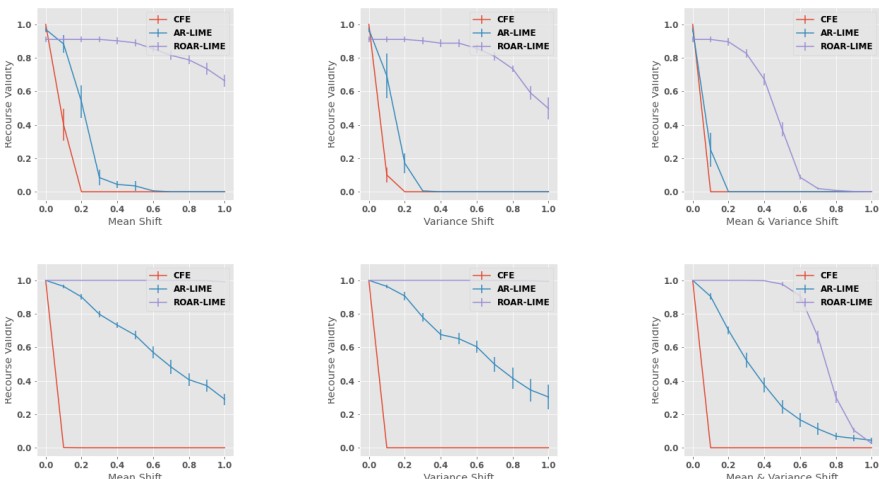

Figure 2: Impact of the degree of data distribution shift on validity of recourse: DNN classifier with $\ell_1$ cost function (top row), DNN classifier with PFC cost function (bottom row); Validity of the recourses generated by all methods drops as degree (magnitude) of the shift increases; The drop in the validity is much smaller for our method ROAR-LIME compared to other baselines.

Here, we assess how different kinds of distribution shifts and the magnitude of these shifts impact the robustness of recourses output by our framework and other baselines. To this end, we leverage our synthetic datasets and introduce mean shifts, variance shifts, and combination shifts (both mean and variance shifts) of different magnitudes by varying $\alpha$ and $\beta$ (See "Synthetic data" in Section 5.1). We then leverage these different kinds of shifted datasets to construct shifted models and then assess the *validity* of the recourses output by our framework and other baselines w.r.t. the shifted models.

We generate recourses using our framework and baselines CFE and AR for different predictive models (LR, DNN) and cost functions ($\ell_1$ distance, PFC). Figure 2 captures the results of this experiment for DNN model both with $\ell_1$ distance and PFC cost functions. Results with other models are included in the Appendix. It can be seen that the x-axis of each of these plots captures the magnitude of the dataset shift, and the y-axis captures the *validity* of the recourses w.r.t. the corresponding shifted model. Standard error bars obtained by averaging the results over 5 runs are also shown.

It can be seen that as the magnitude of the distribution shift increases, *validity* of the recourses generated by all the methods starts dropping. This trend prevailed across mean, variance, and combination (mean and variance) shifts. It can also be seen that the rate at which *validity* of the recourses generated by our method, ROAR-LIME, drops is much smaller compared to that of other baselines CFE and AR-LIME. Furthermore, our method exhibits the highest *validity* compared to the baselines as the magnitude of the distribution shift increases. CFE seems to be the worst performing baseline and the *validity* of the recourses generated by CFE drops very sharply even at small magnitudes of distribution shifts.

## 6 Conclusions & Future Work

We proposed a novel framework, RObust Algorithmic Recourse (ROAR), to address the critical but under-explored issue of recourse robustness to model updates. To this end, we introduced a novel minimax objective to generate recourses that are robust to model shifts, and leveraged adversarial training to optimize this objective. We also presented novel theoretical results which demonstrate that recourses without accounting for model shifts are likely to be invalidated, underscoring the necessity of ROAR. Furthermore, we also showed that the additional cost incurred by robust recourses generated by ROAR are bounded. Extensive experimentation with real world and synthetic datasets demonstrated that recourses using ROAR are highly robust to model shifts induced by a range of data distribution shifts. Our work also paves the way for further research into techniques for generating robust recourses. For instance, it would be valuable to further analyze the tradeoff between recourse robustness and cost to better understand the impacts to affected individuals. Other interesting future directions include non-linear extensions that leverage novel local linear approximation methods that improve on LIME [33].

## Acknowledgements

We would like to thank the anonymous reviewers for their insightful feedback. This work is supported in part by the NSF awards #IIS-2008461 and #IIS-2040989, and research awards from the Harvard Data Science Institute, Amazon, Bayer, and Google. SJ would like to acknowledge the support of the Center for Research on Computation and Society (CRCS) at the Harvard John A. Paulson School of Engineering and Applied Sciences. The views expressed are those of the authors and do not reflect the official policy or position of the funding agencies.

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
