## A Appendix

### A.1 Notation and preliminaries

We consider the metric space $(\mathcal{X}, d(\cdot, \cdot))$ where $d : \mathcal{X} \times \mathcal{X} \to \mathbb{R}_+$. We consider $\ell(\cdot, \cdot)$ to be the cross-entropy loss

$$\ell_{\log}(\mathcal{M}(x), y) \triangleq -y \log \sigma(\mathcal{M}(x)) - (1 - y) \log \sigma((1 - \mathcal{M}(x)))$$

(where $\sigma(x) = \frac{1}{1+\exp(-x)}$ is the sigmoid function) or the $\ell_2$ loss. $c(x, x') : \mathcal{X} \times \mathcal{X} \to \mathbb{R}_+$ is any differentiable cost function as defined in Section 3.1. Assume that $(\mathcal{X}, d)$ has bounded diameter, i.e. $D = \sup_{x,x' \in \mathcal{X}} < \infty$. We restrict to the class of linear models i.e. $\mathcal{M}(x) = w^T x$. W.l.o.g, we assume no bias term. As in Equation 3, the robust models are trained with additive shifts, i.e. $\mathcal{M}_\delta(x) = (w + \delta)^T x$.

Let $x''$ be the recourse obtained as the solution of the proposed objective 4. That is:

$$x'' = \arg\min_{x'' \in \mathcal{A}_x} \max_{\delta \in \Delta} \lambda c(x, x'') + \ell(\mathcal{M}_\delta(x''), 1) \tag{7}$$

Denote

$$\delta = \arg\max_{\delta \in \Delta} \ell(\mathcal{M}_\delta(x''), 1) + \lambda c(x'', x)$$

**Lemma 1.** $\phi(x) = \log\left(1 + \exp\left(-w^T x\right)\right)$ is Lipschitz with $L = \|w\|_2$.

*Proof.*

$$L = \sup_x \|\phi'(x)\|_2 = \sup_x \|-\frac{\exp\left(-w^T x\right)}{1 + \exp\left(-w^T x\right)} w\|_2 = \|w\|_2. \tag{8}$$

$\square$

**Lemma 2.** *van Handel [27] If $(\mathcal{X}, d)$ has bounded diameter $D = \sup_{x,x' \in \mathcal{X}} d(x, x')$, then for any probability measure $\nu$ on $\mathcal{X}$ and 1-Lipschitz function $\phi(x)$ over $(\mathcal{X}, d)$,*

$$\nu\{\phi \geq \mathbb{E}_\nu \phi + \epsilon\} \leq \exp \frac{-2\epsilon^2}{D^2} \tag{9}$$

## B Proof of Theorem 1

**Theorem 1.** *For a given instance $x \sim \mathcal{N}(\mu, \Sigma)$, let $x'$ be the recourse that lies on the original data manifold (conditioned on the event that $\mathcal{M}(x') > 0.5$) and is obtained without accounting for model shifts. Let $\Sigma = UDU^T$. Then, for some* true *model shift $\delta$, such that, $\frac{w^T \mu}{(w+\delta)^T \mu} \geq \frac{\|\sqrt{D}Uw\|}{\|\sqrt{D}U(w+\delta)\|}$, and*

$\sqrt{\frac{2e}{\pi}} \frac{\sqrt{\beta-1}}{\beta} \exp^{-\beta \frac{(w^T \mu)^2}{4\|\sqrt{D}Uw\|^2}} \geq erfc(-\frac{(w+\delta)^T \mu}{\sqrt{2}\|w+\delta\|})$, *for $\beta \geq 1$, the probability that $x'$ is invalidated on $f_{w+\delta}$ is at least: $\frac{1}{2}\sqrt{\frac{2e}{\pi}} \frac{\sqrt{\beta-1}}{\beta} \exp^{-\beta \frac{(w^T \mu)^2}{4\|\sqrt{D}Uw\|^2}} - \frac{1}{2}erfc(-\frac{(w+\delta)^T \mu}{\sqrt{2}\|w+\delta\|})$ where erfc is the complementary gaussian error function*

*Proof.* A $x'$ obtained without accounting for model shifts is valid for $f_w(\cdot)$ is invalidated on the robust classifier $\mathcal{M}_\delta(\cdot)$ if, $x' \in \Omega$ where

$$\Omega = \{x' : w^T x' > 0 \cap (w + \delta)^T x' \leq 0\}$$

Integrating over the set $\Omega$ under the Gaussian pdf, we have:

$$P(x' \text{ is invalidated}) = \frac{1}{(2\pi)^{D/2}\sqrt{|\Sigma|}} \times$$
$$\int_\Omega \exp\left(-\frac{1}{2}(x' - \mu)^T \Sigma^{-1}(x' - \mu)\right) dx \tag{10}$$

Transforming variables s.t. $v = x' - \mu$, and $\Omega_v = \{v\colon w^T(v + \mu) > 0 \cap (w + \delta)^T(v + \mu) \leq 0\}$:

$$P(x' \text{ is invalidated}) = \frac{1}{(2\pi)^{D/2}\sqrt{|\Sigma|}} \int_{\Omega_v} \exp\left(-\frac{1}{2}v^T\Sigma^{-1}v\right)dv \tag{11}$$

Let $\Sigma = UDU^T$, and $z = Uv$, the transformed hyperplanes are given by: $w_1 = Uw$ i.e. $w = U^Tw_1$. Similarly, Let $U\delta = \delta_1$, then $U(w + \delta) = Uw + U\delta = w_1 + \delta_1$ i.e., $(w + \delta) = U^T(w_1 + \delta_1)$. Then $\Omega_z = \{z\colon w_1^Tz + w^T\mu > 0 \cap (w_1 + \delta_1)^Tz + (w + \delta)^T\mu \leq 0\}$ and:

$$P(x' \text{ is invalidated}) = \frac{1}{(2\pi)^{D/2}\sqrt{|\Sigma|}} \int_{\Omega_z} \exp\left(-\frac{1}{2}z^TD^{-1}z\right)dz \tag{12}$$

Finally, transforming $z$ such that, $t = \sqrt{D}^{-1}z$ or $z = \sqrt{D}t$ $w_2 = \sqrt{D}w_1$ and $\delta_2 = \sqrt{D}\delta_1$, we have,

$$\Omega_t = \{t\colon w_2^Tt + w^T\mu > 0 \cap (w_2 + \delta_2)^Tt + (w + \delta)^T\mu \leq 0\}$$

. Therefore:

$$P(x' \text{ is invalidated}) = \frac{1}{(2\pi)^{d/2}} \int_{\Omega_t} \exp\left(-\frac{1}{2}t^Tt\right)dt \tag{13}$$

Finally, let $P$ be an orthogonal projection matrix s.t. $Pw_2 = \|w_2\|e_d$ where $e_d$ is the basis vector for dimension $d$. By definition of the projection matrix in 1-d, we have $P = \frac{e_de_d^T}{\|e_d\|^2} = e_de_d^T$. Thus the projection $P(w_2 + \delta_2) = \|w_2\|e_d + e_de_d^T\delta_2 = \|w_2 + \delta_2\|e_d$. Transforming $t$ s.t. $b = Pt$, we have $\Omega_b = \{b\colon b^T(\|w_2\|e_d) + w^T\mu > 0 \cap b^T(\|w_2 + \delta_2\|e_d) + (w + \delta)^T\mu \leq 0\}$

$$P(x \text{ is invalidated}) = \frac{1}{\sqrt{(2\pi)}} \int_{\Omega_b} \exp\left(-\frac{1}{2}b^Tb\right)db \tag{14}$$

Simplifying $\Omega_b$, we have that:

$$\Omega_b = \{b\colon \mathbb{R}^{D-1} \times [c_1, c_2]\}$$

s.t. $c_1 = \frac{-w^T\mu}{\|w_2\|}$ and $c_2 = \frac{-(w+\delta)^T\mu}{\|w_2 + \delta_2\|}$ and:

$$P(x \text{ is invalidated}) = \frac{1}{\sqrt{(2\pi)}} \int_{c_1}^{c_2} \exp\left(-\frac{1}{2}s^2\right)ds \tag{15}$$

We restrict to the case of $c_1 \leq c_2$ i.e. :

$$\frac{w^T\mu}{(w + \delta)^T\mu} \geq \frac{\|w_2\|}{\|w_2 + \delta_2\|} = \frac{\|\sqrt{D}Uw\|}{\|\sqrt{D}U(w + \delta)\|} \tag{16}$$

If $c_1 > c_2$, this implies that the true shift $\delta$ is such that non-robust recourse not invalid. We bound Eq. 15 using the Gaussian Error Function defined as follows:

$$\text{erf}(z) = \frac{2}{\sqrt{\pi}} \int_0^z \exp{-t^2}dt \tag{17}$$

and the complementary error function as: erfc(z) $= 1 - $ erf(z).

From Glaisher [9], we have that:

$$\left(\frac{c}{\pi}\right)^{1/2} \int_p^q \exp{(-cx^2)}dx = \frac{1}{2}(\text{erf}(q\sqrt{c}) - \text{erf}(p\sqrt{c})) \tag{18}$$

with $c = \frac{1}{2}$, we have,

$$P(x' \text{ is invalidated}) = \frac{1}{2}(\text{erf}(c_2/\sqrt{2}) - \text{erf}(c_1/\sqrt{2}))$$
$$= \frac{1}{2}(\text{erfc}(c_1/\sqrt{2}) - \text{erfc}(c_2/\sqrt{2})) \tag{19}$$

From Chang et al. [5], we note the following upper and lower bounds of the error function:

$$\text{erfc}(c) \leq \exp^{-c^2/2} \tag{20}$$

$$\text{erfc}(c) \geq \sqrt{\frac{2e}{\pi}} \frac{\sqrt{\beta - 1}}{\beta} \exp^{-\beta c^2/2} \tag{21}$$

where $\beta \geq 1$.

Then if the condition $\sqrt{\frac{2e}{\pi}} \frac{\sqrt{\beta-1}}{\beta} \exp^{-\beta \frac{(w^T \mu)^2}{4\|\sqrt{D}Uw\|^2}} \geq \text{erfc}(-\frac{(w+\delta)^T \mu}{\sqrt{2}\|w+\delta\|})$ is satisfied, mainly to confirm that the lower bound on the left side still dominates the second term, we have a lower bound given by:

$$
\begin{aligned}
P(x' \text{ is invalidated}) &= \frac{1}{2}(\text{erf}(c_2/\sqrt{2}) - \text{erf}(c_1/\sqrt{2})) \\
&\geq \frac{1}{2}\sqrt{\frac{2e}{\pi}} \frac{\sqrt{\beta-1}}{\beta} \exp^{-\beta c_1^2/4} - \frac{1}{2}\text{erfc}(c_2/\sqrt{2}) \\
&= \frac{1}{2}\sqrt{\frac{2e}{\pi}} \frac{\sqrt{\beta-1}}{\beta} \exp^{-\beta \frac{(w^T \mu)^2}{4\|\sqrt{D}Uw\|^2}} - \frac{1}{2}\text{erfc}(-\frac{(w+\delta)^T \mu}{\sqrt{2}\|w+\delta\|})
\end{aligned}
\tag{22}
$$

We notice that for any constant, $\exists \beta$ s.t. the RHS is maximized. An optimal $\beta$ can be found for every example in this manner. For this bound, $\beta_{opt} = 1.080$ when $x$ is normally distributed. $\qquad \square$

## B.1 Discussion on other distributions

Here we give illustrations of how recourses can be invalidated for other distributions like Bernoulli, Uniform, and Categorical.

**Remark 1.** *Let $x \sim Bernoulli(p)$, where $0 \leq p \leq 1$. To bound the probability that a recourse provided for samples from this distribution is invalidated due to model shifts, we observe the following. Let the classifier for such samples be given by a threshold $\tau$ where $0 < \tau < 1$ that is, $y = \mathbf{1}(x > \tau)$. Then recourse is provided for all samples where $x = 0$ and is given by $x' = 1$. Now consider model shifts $\delta$ to $\tau$. Then, we have that for all $\delta$ such that $\tau + \delta < 1$, $p(x' \text{is invalidated}) = 0$. On the other hand, for all $\delta$ such that $\tau + \delta \geq 1$, $p(x' \text{is invalidated}) = 1$ and in fact no sample is favorably classified ($y = 1$ is favorable).*

**Remark 2.** *Let $x \sim Unif(a, b)$. The argument for Uniform distribution follows similarly. Let the classifier for such samples be given by a threshold $\tau$ where $a < \tau < b$ that is, $y = \mathbf{1}(x > \tau)$. Then recourse is provided for all samples where $x < \tau$ and is given by $x' > \tau$. Now consider model shifts $\delta$ to $\tau$. Then, we have that:*

$$
p(x' \text{ is invalidated }) = \begin{cases} 0 & \text{for } \delta \leq a - \tau \\ \frac{\delta}{b-a}, & \text{for } a - \tau \leq \delta \leq b - \tau \\ 1 & \text{for } \delta \geq b - \tau \end{cases}
$$

**Remark 3.** *Let $k \sim Categorical(p, K)$ where $p \in \Delta^{K-1}$. For simplicity let $[K] = \{1, 2, \cdots, K\}$. We motivate this remark for a classifier $\mathbf{w}$ where $y = \mathbf{1}(\mathbf{w}^T \mathbf{x} > \tau)$ where $\mathbf{x}$ is the one-hot encoded version of $k$ i.e. $\mathbf{x}_k = 1$ and is $\mathbf{x}_{[K] \setminus k} = 0$. Since only one of the elements of $\mathbf{x}$ is $1$ at any time, it is clear that $y = 0$ for all $k$ such that $\kappa_0 = \{k : k \in [K] \text{ and } w_k \leq \tau\}$ and $y = 1$ for $\kappa_1 = \{k : k \in [K] \text{ and } w_k > \tau\}$. Assume that flipping to any category from the current category is equally costly. Then for all samples s.t. $k \in \kappa_0$, the recourse provided is any one category randomly chosen from $\kappa_1$. As before let $\delta$ be the vector representing model shift to the original model parametrized by $\mathbf{w}$. Then, with the same threshold $\tau$, under this model, $\kappa_0^\delta = \{k : k \in [K] \text{ and } w_k + \delta_k \leq \tau\}$. Then the probability of a recourse being invalidated is: $\frac{|\kappa_0^\delta \cap \kappa_1|}{|\kappa_1|}$*

Generalizations to multivariate families like exponential family distributions is left to future work.

## C   Proof of Theorem 2

**Theorem 2.** *Let $x \in \mathcal{X}$, and $x \sim \nu$ where $\nu$ is a distribution such that $\mathbb{E}_\nu[x] = \mu < \infty$, where $\mathcal{X}$ is a metric space $(\mathcal{X}, d(\cdot, \cdot))$ and $d : \mathcal{X} \times \mathcal{X} \to \mathbb{R}_+$. Let $d \triangleq \ell_2$ and assume that $(\mathcal{X}, d)$ has bounded diameter $D = \sup_{x, x' \in \mathcal{X}} d(x, x')$. Let recourses obtained without accounting for model shifts and constrained to the manifold be denoted by $x' \sim \nu$, and robust recourses be denoted by $x''$. Let $\delta > 0$ be the shift that maximizes Eq. 3 for sample $x$ corresponding to $x''$. Further assume that the ROAR objective (Equation 3) is convex in $x''$ for a fixed $\delta$. For $\ell \triangleq \ell_{log}$ (the cross-entropy loss), some $0 < \eta' \ll 1$, w.h.p. $(1 - \eta')$, we have that:*

$$c(x'', x) - c(x', x) \leq \frac{1}{\lambda} \frac{1}{\|w + \delta\|} \mathbb{E}_\nu[\exp -\phi(w + \delta)^T x'] + \sqrt{\frac{D^2}{2} \log \left(\frac{1}{\eta'}\right)}\right) \tag{23}$$

*Proof.* Since $x''$ is the minimizer of Equation 7 and using convexity in $x'$ for a fixed $\delta$,

$$\frac{1}{\|w + \delta\|} \ell_{log}(\mathcal{M}_\delta(x''), 1) + \lambda c(x'', x) \leq \tag{24}$$

$$\frac{1}{\|w + \delta\|} \ell_{log}(\mathcal{M}_\delta(x'), 1) + \lambda c(x', x) \tag{25}$$

$$c(x'', x) - c(x', x) \leq \frac{(\ell_{log}(\mathcal{M}_\delta(x'), 1) - \ell_{log}(\mathcal{M}_\delta(x''), 1))}{\lambda \|w + \delta\|} \tag{26}$$

$$= \frac{1}{\lambda \|w + \delta\|} \left\{ -\log \sigma(w + \delta)^T x' \right. \tag{27}$$

$$\left. + \log \sigma(w + \delta)^T x'' \right\} \tag{28}$$

$$= \frac{1}{\lambda \|w + \delta\|} \log \left\{ \frac{1 + \exp -(w + \delta)^T x'}{1 + \exp -(w + \delta)^T x''} \right\} \tag{29}$$

$$= \frac{1}{\lambda \|w + \delta\|} \left( \log \left\{ 1 + \exp -(w + \delta)^T x' \right\} \right. \tag{30}$$

$$\left. - \log \left\{ 1 + \exp -(w + \delta)^T x'' \right\} \right) \tag{31}$$

$$\leq \frac{1}{\lambda \|w + \delta\|} \log \left\{ 1 + \exp -(w + \delta)^T x' \right\} \tag{32}$$

from re-arranging, and where the last bound comes from the fact that $\log \left\{ 1 + \exp^{-(w+\delta)^T x'} \right\} \geq 0$.

From Lemma 1, we know that $\frac{1}{\|w+\delta\|} \log \left\{ 1 + \exp^{-(w+\delta)^T x'} \right\}$ is 1-Lipschitz. Therefore we can apply the upper bound from Lemma 2, to Eq. 32. This results in the following, w.h.p:

$$c(x'', x) - c(x', x) \leq \frac{1}{\lambda} \frac{1}{\|w + \delta\|} \mathbb{E}_\nu[\log \left\{ 1 + \exp -(w + \delta)^T x' \right\}] + \sqrt{\frac{D^2}{2} \log \left(\frac{1}{\eta'}\right)}\right) \tag{33}$$

We now bound the expectation $\mathbb{E}_\nu[\log \left\{ 1 + \exp -(w + \delta)^T x' \right\}] \triangleq \mathbb{E}_\nu[\phi((w + \delta)^T x')]$.

Analytical expressions and/or approximations for $\mathbb{E}_\nu \phi(w^T x')$ i.e. the logistic function are generally not available in closed form. For simplification, we bound $\mathbb{E}_\nu[\log (1 + \exp^{-z})]$. Consider the case

where $\nu \triangleq \mathcal{N}(\mu, \sigma^2)$. Then we have:

$$\mathbb{E}_\nu[\log\left(1 + \exp^{-z}\right)] = \mathbb{E}_\nu[\frac{\ln\left(1 + \exp^{-z}\right)}{\ln 2}]$$

$$\leq \frac{1}{\ln 2}\mathbb{E}_\nu[\exp{-z}]$$

$$= \frac{\sqrt{2\pi\sigma^2}}{\ln 2}\frac{1}{\sqrt{2\pi\sigma^2}}\mathbb{E}_\nu[\exp{-z}]$$

$$= \frac{\sqrt{2\pi\sigma^2}}{\ln 2}\frac{1}{\sqrt{2\pi\sigma^2}}\int_x \exp{-x}\exp{-\frac{(x-\mu)^2}{2\sigma^2}}dx$$

$$= \frac{\sqrt{2\pi\sigma^2}}{\ln 2}\frac{1}{\sqrt{2\pi\sigma^2}}\int_x \exp{\frac{[-x^2 + 2\mu x - \mu^2 - 2\sigma^2 x]}{2\sigma^2}}dx$$

$$= \frac{\sqrt{2\pi\sigma^2}}{\ln 2}\frac{1}{\sqrt{2\pi\sigma^2}}\int_x \exp{\frac{1}{2\sigma^2}[-x^2 + 2\mu x - 2\sigma^2 x - \mu^2 + (\mu - \sigma^2)^2 - (\mu - \sigma^2)^2]}dx$$

$$= \frac{\sqrt{2\pi\sigma^2}}{\ln 2}\frac{1}{\sqrt{2\pi\sigma^2}}\int_x \exp{\frac{1}{2\sigma^2}[-x^2 + 2(\mu - \sigma^2)x - (\mu - \sigma^2)^2]}\exp{[-\mu^2 + (\mu - \sigma^2)^2]}dx$$

$$= \frac{\sqrt{2\pi\sigma^2}}{\ln 2}\frac{1}{\sqrt{2\pi\sigma^2}}\exp{\frac{[(\mu - \sigma^2)^2 - \mu^2]}{2\sigma^2}}\int_x \exp{\frac{1}{2\sigma^2}}-[x^2 - 2(\mu - \sigma^2)x + (\mu - \sigma^2)^2]dx$$

$$= \frac{\sqrt{2\pi\sigma^2}}{\ln 2}\exp{\frac{(\mu - \sigma^2 - \mu)(\mu - \sigma^2 + \mu)}{2\sigma^2}}\frac{1}{\sqrt{2\pi\sigma^2}}\int_x \exp{\frac{1}{2\sigma^2}}-[x - (\mu - \sigma^2)]^2 dx$$

$$= \frac{\sqrt{2\pi\sigma^2}}{\ln 2}\exp{(\mu - \frac{\sigma^2}{2})}$$

$$\tag{34}$$

Thus for Gaussian distributed case, w.h.p,

$$c(x'', x) - c(x', x) \leq \frac{1}{\lambda}\frac{1}{\|w + \delta\|}\frac{\sqrt{2\pi\sigma^2}}{\ln 2}\exp{(\mu - \frac{\sigma^2}{2})} + \sqrt{\frac{D^2}{2}\log{(\frac{1}{\eta'})}}\right) \tag{35}$$

$\square$

Interpreting the bound, we see that the bound is significantly loose $\mu \ll 0$ if and improves when $\mu > 0$. Note that here, the bound cannot be improved significantly by conditioning on the event that $f(x') > 0.5$ or equivalently that $w^T x' > 0$, since we are concerned with bounding a quantity that is a function of the shifted decision boundary $w + \delta$. We have also added these interpretations as you suggested.

## D Experiments

### D.1 Experimental setup

All experiments were run on a 2 GHz Quad-Core Intel Core i5.

**Real world datasets**  Below we provide a complete list of all the features we used in each of our datasets.

The features we use for the German credit dataset are: "duration", "amount", "age", "personal_status_sex".

The features we use for the SBA Case dataset (temporal shift) are: 'Zip', 'NAICS', 'ApprovalDate', 'ApprovalFY', 'Term', 'NoEmp', 'NewExist', 'CreateJob', 'RetainedJob', 'FranchiseCode', 'UrbanRural', 'RevLineCr', 'ChgOffDate', 'DisbursementDate', 'DisbursementGross', 'ChgOffPrinGr', 'GrAppv', 'SBA_Appv', 'New', 'RealEstate', 'Portion', 'Recession', 'daysterm', 'xx'.

The features we use for the Student Performance dataset (geospatial shift) are: 'sex', 'age', 'address', 'famsize', 'Pstatus', 'Medu', 'Fedu', 'Mjob', 'Fjob', 'reason', 'guardian', 'traveltime', 'studytime',

'failures', 'schoolsup', 'famsup', 'paid', 'activities', 'nursery', 'higher', 'internet', 'romantic', 'famrel', 'freetime', 'goout', 'Dalc', 'Walc', 'health', 'absences'.

**Predictive models**   We use a 3-layer DNN with 50, 100, and 200 nodes in each consecutive layer. We use ReLU activation, binary cross entropy loss, adam optimizer, and 100 training epochs.

**Cost functions**   For PFC, we simulate pairwise feature preferences with 200 comparisons per feature pair with preferences assigned randomly. After passing these preferences to the Bradley-Terry model, we shift the output by its minimum so that all feature costs are non-negative.

**Setting and implementation details**

**Real world experiments.**   Recall we partition our data into initial data , $D_1$, and shifted data, $D_2$. For the German credit dataset (correction shift) we use the original version as $D_1$ and the corrected version as $D_2$. For the SBA case dataset (temporal shift) we use the data from 1989-2006 as $D_1$ and all the data from 1986-2012 for $D_2$. For the student performance dataset (geospatial shift) we use the data from GP for $D_1$ and the data from both schools, GP and MS, for $D_2$.

For our experiments on real world data we use 5-fold cross validation. In each trial, we train $\mathcal{M}_1$ on 4 folds of $D_1$ and $\mathcal{M}_2$ on 4 folds of $D_2$. Average $\mathcal{M}_1$ and $\mathcal{M}_2$ accuracy and AUC on the remaining fold (hold out set) of $D_1$ and $D_2$ respectively, is reported in Table 2. We then find recourses on the remaining $D_1$ fold (hold out set). Results on average $\mathcal{M}_1$ validity, $\mathcal{M}_2$ validity, and recourse costs across the 5 trials are included in Table 1 in Section 5 of the main paper.

| | | | Accuracy | AUC |
|---|---|---|---|---|
| Correction | LR | $\mathcal{M}_1$ | $0.70 \pm 0.01$ | $0.65 \pm 0.01$ |
| | | $\mathcal{M}_2$ | $0.71 \pm 0.02$ | $0.65 \pm 0.02$ |
| | NN | $\mathcal{M}_1$ | $0.71 \pm 0.02$ | $0.66 \pm 0.02$ |
| | | $\mathcal{M}_2$ | $0.69 \pm 0.03$ | $0.67 \pm 0.03$ |
| | SVM | $\mathcal{M}_1$ | $0.71 \pm 0.01$ | $0.65 \pm 0.01$ |
| | | $\mathcal{M}_2$ | $0.71 \pm 0.02$ | $0.65 \pm 0.02$ |
| Temporal | LR | $\mathcal{M}_1$ | $1.00 \pm 0.00$ | $1.00 \pm 0.00$ |
| | | $\mathcal{M}_2$ | $0.99 \pm 0.00$ | $1.00 \pm 0.00$ |
| | NN | $\mathcal{M}_1$ | $1.00 \pm 0.00$ | $1.00 \pm 0.00$ |
| | | $\mathcal{M}_2$ | $0.99 \pm 0.00$ | $1.00 \pm 0.00$ |
| | SVM | $\mathcal{M}_1$ | $1.00 \pm 0.00$ | $1.00 \pm 0.00$ |
| | | $\mathcal{M}_2$ | $0.99 \pm 0.00$ | $1.00 \pm 0.00$ |
| Geospatial | LR | $\mathcal{M}_1$ | $0.92 \pm 0.01$ | $0.77 \pm 0.06$ |
| | | $\mathcal{M}_2$ | $0.85 \pm 0.02$ | $0.73 \pm 0.02$ |
| | NN | $\mathcal{M}_1$ | $0.94 \pm 0.01$ | $0.72 \pm 0.06$ |
| | | $\mathcal{M}_2$ | $0.84 \pm 0.03$ | $0.70 \pm 0.03$ |
| | SVM | $\mathcal{M}_1$ | $0.92 \pm 0.01$ | $0.75 \pm 0.07$ |
| | | $\mathcal{M}_2$ | $0.85 \pm 0.02$ | $0.73 \pm 0.02$ |

Table 2: Average test accuracy and AUC of LR, NN, and SVM models on real world datasets

**Synthetic experiments.**   Our synthetic experiment setting mirrors our real world experiment setting. We partition $D_1$ (original data) and $D_2$ (mean and/or variance shifted data) into 5 folds, training $\mathcal{M}_1$ on 4 folds of $D_1$ and $\mathcal{M}_2$ on 4 folds of $D_2$. We then find recourse on the remaining $D_1$ fold (hold out set).

**Nonlinear predictive models.**   For AR and ROAR, we use LIME to find local linear approximations when the underlying predictive models are non-linear. We use the default implementation of LIME by Ribeiro et al. [25] with logistic regression as the local linear model class.

**D.2   Additional Experimental Results**

We include in this Appendix additional empirical results that were omitted from Section 5 of the main paper due to space constraints:

- We show the Avg. Cost, $\mathcal{M}_1$ Validity, and $\mathcal{M}_2$ Validity of recourses across different real world datasets with SVM predictive model (Table 4). Results for other predictive models are included in Table 1 in Section 5 of the main paper.

- We compute the Theorem 2 upper bounds on ROAR cost and compare them to our empirical results in Table 3.
- We show the impact of the degree of mean, variance, combination (both mean and variance) distrbution shifts on the validity of recourses for LR and SVM models under different cost functions (Figure 3). Results with DNN model are included in Figure 2 in Section 5 of the main paper.
- We compute the avg. cost (both $\ell_1$ distance and PFC) of recourses corresponding to LR, DNN, SVM models on synthetic datasets. Results are presented in Table 5.
- We fix $\lambda = 0.1$ and evaluate the effect of $\delta_{max}$ on the validity and avg cost of ROAR recourses. Results for correction, temporal, and geospatial shift are shown in Figures 4, 5, and 6 respectively.
- We fix $\lambda = 0.1$ and evaluate the effect of $\delta_{max}$ on the validity and avg cost of ROAR-MINT recourses. Results are reported in Figure 7.

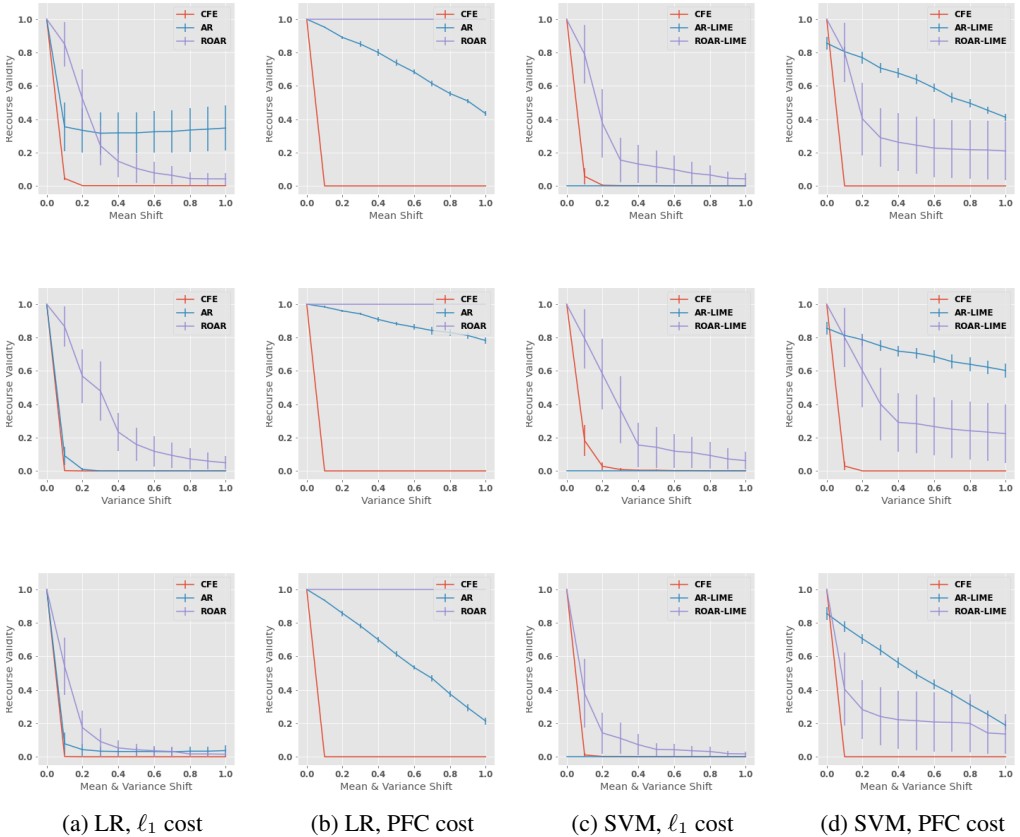

(a) LR, $\ell_1$ cost  (b) LR, PFC cost  (c) SVM, $\ell_1$ cost  (d) SVM, PFC cost

Figure 3: Impact of the degree of mean (top row), variance (middle row), and combination (bottom row) distribution shift on validity of recourse with LR and SVM models. Validity of the recourses generated by all methods drops as degree (magnitude) of the shift increases; In the majority of settings, ROAR is the most robust, maintaining higher validity as shift increases compared to other baselines. Results with DNN model are shown in Figure 2 in Section 5 of the main paper.

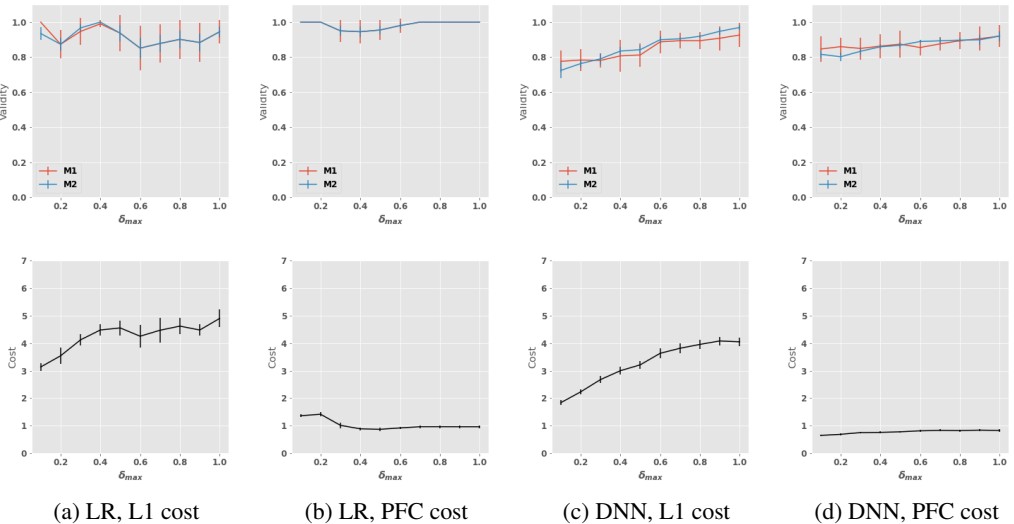

(a) LR, L1 cost      (b) LR, PFC cost      (c) DNN, L1 cost      (d) DNN, PFC cost

Figure 4: ROAR $\mathcal{M}_1$ (original model) and $\mathcal{M}_2$ (shifted model) Validity (top) and Avg Cost (bottom) for different values of $\delta_{max}$ on the German credit dataset (correction shift). Notice that as $\delta_{max}$ increases, ROAR remains robust (high $\mathcal{M}_2$ validity), but L1 cost increases.

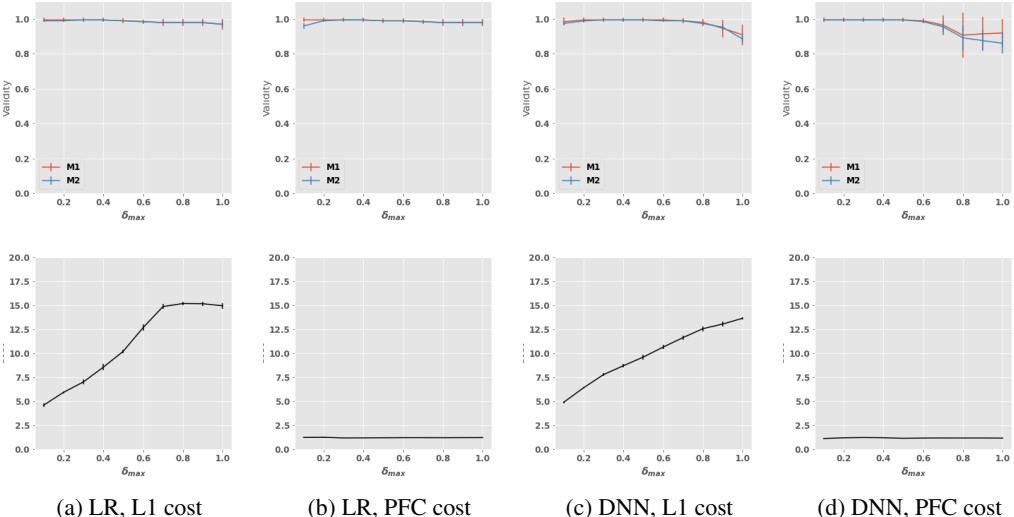

(a) LR, L1 cost      (b) LR, PFC cost      (c) DNN, L1 cost      (d) DNN, PFC cost

Figure 5: ROAR $\mathcal{M}_1$ (original model) and $\mathcal{M}_2$ (shifted model) Validity (top) and Avg Cost (bottom) for different values of $\delta_{max}$ on the SBA case dataset (temporal shift). Notice that as $\delta_{max}$ increases, ROAR remains robust (high $\mathcal{M}_2$ validity), but L1 cost increases.

|  |  |  | Recourse Cost | |
| Model | Cost | Data | Empirical Estimate | Theoretical Upper Bound |
|---|---|---|---|---|
|  |  | Correction | $3.13 \pm 0.32$ | $4.61 \pm 0.45$ |
|  | L1 | Temporal | $3.14 \pm 0.25$ | $51.45 \pm 31.26$ |
| LR |  | Geospatial | $10.88 \pm 1.67$ | $36.09 \pm 1.37$ |
|  |  | Correction | $0.36 \pm 0.08$ | $0.26 \pm 0.11$ |
|  | PFC | Temporal | $0.44 \pm 0.12$ | $3.40 \pm 2.80$ |
|  |  | Geospatial | $1.20 \pm 0.10$ | $1.35 \pm 0.18$ |

Table 3: The empirical estimate refers to the cost of ROAR, and is reproduced from Table 1 in the main paper. The theoretical upper bound is computed analytically with respect to CFE cost with $\eta = 0.05$. Though our real world experiments do not satisfy the Theorem 2 Gaussian and manifold assumptions, in the majority of cases the empirical estimate falls below the theoretical upper bound, aligning with our Theorem 2 result.

| Model | Cost | Recourse | Correction Shift | | | Temporal Shift | | | Geospatial Shift | | |
|---|---|---|---|---|---|---|---|---|---|---|---|
|  |  |  | Avg Cost | $\mathcal{M}_1$ Validity | $\mathcal{M}_2$ Validity | Avg Cost | $\mathcal{M}_1$ Validity | $\mathcal{M}_2$ Validity | Avg Cost | $\mathcal{M}_1$ Validity | $\mathcal{M}_2$ Validity |
|  |  | CFE | $0.83 \pm 0.12$ | $1.00 \pm 0.00$ | $0.54 \pm 0.30$ | $3.53 \pm 0.45$ | $1.00 \pm 0.00$ | $0.38 \pm 0.12$ | $5.96 \pm 0.46$ | $1.00 \pm 0.00$ | $0.11 \pm 0.05$ |
|  |  | AR | $0.85 \pm 0.12$ | $1.00 \pm 0.00$ | $0.54 \pm 0.30$ | $1.85 \pm 0.30$ | $1.00 \pm 0.00$ | $0.57 \pm 0.13$ | $4.03 \pm 0.17$ | $0.06 \pm 0.04$ | $0.23 \pm 0.09$ |
|  | L1 | ROAR | $3.57 \pm 0.33$ | $0.84 \pm 0.08$ | $0.87 \pm 0.07$ | $4.66 \pm 0.71$ | $0.99 \pm 0.01$ | $\mathbf{0.98 \pm 0.01}$ | $15.12 \pm 1.56$ | $1.00 \pm 0.00$ | $\mathbf{0.92 \pm 0.14}$ |
|  |  | MINT | $4.90 \pm 0.69$ | $1.00 \pm 0.00$ | $0.90 \pm 0.12$ | NA | NA | NA | NA | NA | NA |
| SVM |  | ROAR-MINT | $3.76 \pm 0.14$ | $1.00 \pm 0.00$ | $\mathbf{1.00 \pm 0.00}$ | NA | NA | NA | NA | NA | NA |
|  |  | CFE | $0.07 \pm 0.04$ | $1.00 \pm 0.00$ | $0.48 \pm 0.28$ | $0.23 \pm 0.09$ | $1.00 \pm 0.00$ | $0.36 \pm 0.15$ | $0.25 \pm 0.06$ | $1.00 \pm 0.00$ | $0.10 \pm 0.06$ |
|  |  | AR | $0.09 \pm 0.02$ | $1.00 \pm 0.00$ | $0.65 \pm 0.27$ | $0.13 \pm 0.05$ | $1.00 \pm 0.00$ | $0.56 \pm 0.26$ | $0.20 \pm 0.03$ | $0.14 \pm 0.09$ | $0.11 \pm 0.05$ |
|  | PFC | ROAR | $0.86 \pm 0.29$ | $1.00 \pm 0.00$ | $1.00 \pm 0.00$ | $0.81 \pm 0.27$ | $0.99 \pm 0.01$ | $\mathbf{0.98 \pm 0.01}$ | $1.31 \pm 0.13$ | $1.00 \pm 0.00$ | $\mathbf{0.98 \pm 0.04}$ |
|  |  | MINT | $0.43 \pm 0.03$ | $1.00 \pm 0.00$ | $0.90 \pm 0.12$ | NA | NA | NA | NA | NA | NA |
|  |  | ROAR-MINT | $0.70 \pm 0.03$ | $1.00 \pm 0.00$ | $\mathbf{1.00 \pm 0.00}$ | NA | NA | NA | NA | NA | NA |

Table 4: Avg. Cost, $\mathcal{M}_1$ Validity, and $\mathcal{M}_2$ Validity of recourses across different real world datasets with SVM predictive model. Recourses using our framework (ROAR and ROAR-MINT) are more robust (higher $\mathcal{M}_2$ validity) compared to those generated by existing baselines. Results with other predictive models are shown in Table 1 in Section 5 of the main paper.

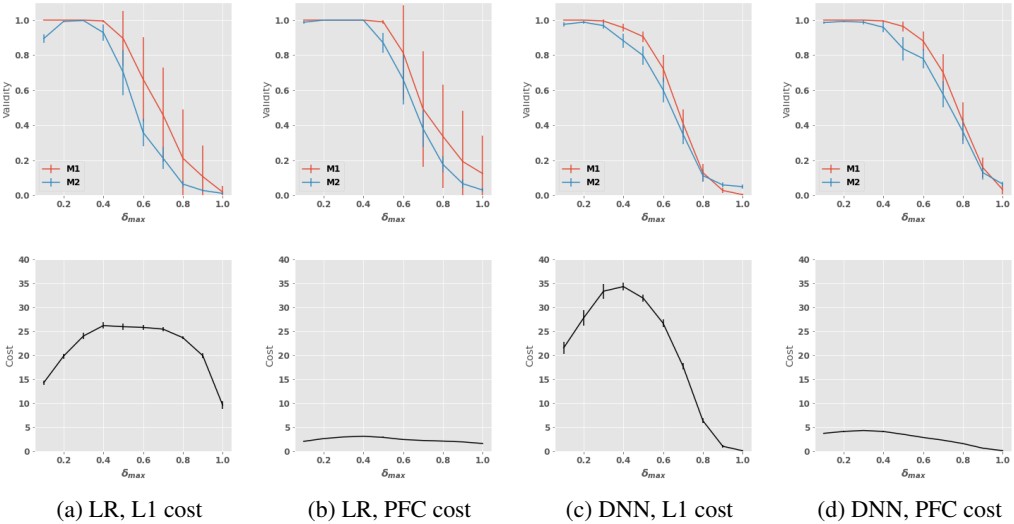

(a) LR, L1 cost        (b) LR, PFC cost        (c) DNN, L1 cost        (d) DNN, PFC cost

Figure 6: ROAR $\mathcal{M}_1$ (original model) and $\mathcal{M}_2$ (shifted model) Validity (top) and Avg Cost (bottom) for different values of $\delta_{max}$ on the student performance dataset (geospatial shift). Notice that validity decreases for $\delta_{max} > 0.4$. This suggests that $\delta_{max} > 0.4$ models a greater shift than the true $\mathcal{M}_1$ to $\mathcal{M}_2$ shift on this dataset.

| | | Avg. Cost of Recourse | | |
|---|---|---|---|---|
| Model | Cost | CFE | AR | ROAR |
| LR | L1 | $3.93 \pm 0.04$ | $3.87 \pm 0.04$ | $4.03 \pm 0.06$ |
| | PFC | $0.11 \pm 0.00$ | $0.00 \pm 0.00$ | $0.89 \pm 0.01$ |
| DNN | L1 | $3.85 \pm 0.06$ | $3.81 \pm 0.05$ | $3.69 \pm 0.12$ |
| | PFC | $0.09 \pm 0.01$ | $0.00 \pm 0.00$ | $0.91 \pm 0.01$ |
| SVM | L1 | $3.92 \pm 0.04$ | $3.49 \pm 0.10$ | $4.01 \pm 0.08$ |
| | PFC | $0.14 \pm 0.01$ | $0.01 \pm 0.00$ | $0.89 \pm 0.01$ |

Table 5: Avg. Cost of recourses on synthetic dataset. Notice that ROAR costs are slightly higher than the baselines, confirming the Theorem 2 result. But on the flip side, ROAR achieves much higher robustness than the baselines, as shown in Figure 2 in Section 5 of the main paper and in Figure 3 in the Appendix for the synthetic dataset.

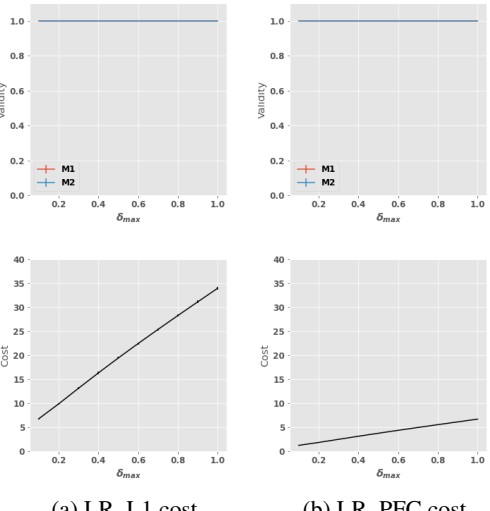

(a) LR, L1 cost          (b) LR, PFC cost

Figure 7: ROAR-MINT $\mathcal{M}_1$ (original model) and $\mathcal{M}_2$ (shifted model) Validity (top) and Avg Cost (bottom) for different values of $\delta_{max}$ on the German credit dataset (correction shift). Notice that as $\delta_{max}$ increases, ROAR-MINT remains robust (high $\mathcal{M}_2$ validity), but cost increases.