# OpenReview forum: "Towards Robust and Reliable Algorithmic Recourse"
_NeurIPS.cc/2021/Conference — NeurIPS 2021 Poster_

### Official Review · Reviewer_efp6 · 2021-07-15

**Rating:** 6
**Confidence:** 4

**Summary:**

The paper proposes a method to construct recourses that are valid under worst case additive model shifts for linear models and applied to non-linear ones with LIME. The paper analyzes these robust recourses theoretically but not for the given algorithm. I think the paper overall is of great quality but it lacks in novelty in particular because of related work that shares the exact same problem formulation and experimental setup (see [25] as authors cite).  The algorithmic recourse problem is also of limited practical applicability, and thinking about robust (but much less realistic) recourses, does not help address the problems that help make algorithmic recourse more practically applicable. The proposed algorithm to solve the problem which is the main contribution of the paper is not analyzed theoretically, and empirical evidence do not discuss Theorem 2 which talks about the average cost of the recourses.
I think the paper is borderline primarily because of how similar it is to prior work.

**Limitations And Societal Impact:**

There is limited discussion to how realistic the recourses suggested in the paper can be applied in practice. There is limited proof that if we take a recourse from any of the datasets, then that recourse establishes a route for an individual to change their discussion, or how can they accomplish that change.

**Main Review:**

Originality: The paper proposes a method to construct recourses that are valid under additive model shifts for linear models and applied to non-linear ones with LIME. The observation that recourses are invalidated with model shifts has been deeply explored in prior work that the authors reference heavily, moreover, the formulation of the problem and the experimental setup mimic exactly that of the related work [25]. Thus the main novelty here is the contribution of a method and worst-case formulation to obtain recourses. They also provide theoretical analysis for these robust recourses (but crucially not those obtained by the algorithm). I think there is limited novelty in the paper in terms of formulation and experiments and the nature of the approach, but it is an interesting combination of ideas.

Quality: The experimental results are clear and very well compared to the baselines and well as the theoretical results are well argued for. However I have some important notes on Theorem 1 which caused a lot of confusion for me, notably how x' is sampled independently. Furthermore, I am a bit uneasy about the increase in avg cost in the experiments and the lack of experimental discussion about it. See comments below for more details.

Clarity: The paper is very well written and easy to follow.

Significance: I think the lesson that recourses become invalidated when models change has already been discussed, so this paper tries to solve that problem with a worst case approach and propose a straightforward gradient method. I haven't seen practical uses for algorithmic recourse yet, nor are any cited (please prove me wrong on this point if I am mistaken). So deriving robust recourses with this approach that has no stated guarantees does not address practical or theoretical challenges that need to be solved first. I am not convinced from the point of average cost that the robustness guaranteed is worth future validity, from the empirical point of view, from the theoretical perspective Th2 is a nice guarantee. The empirical setup is also borrowed from prior work, so that researchers can look at older papers for this kind of setup.

Limitations: 1) no guarantees on algorithm, 2) no empirical discussion of Theorem 2 and the increase in average cost, 3) limited form of model shifts that apply only to linear models.

Comments:

Extension to multiclass?

Can we define recourses in terms of probabilistic model? I.e. a recourse of \delta is that that increases the probability of a positive outcome by \delta, or sequential recourses that can be thought of as a curriculum for the individual.  That I think is a much more useful practical definition.

Supposedly as designers we control the model and have historical data on our users and recourses, why not update the model constrained by the fact that old recourses must still hold? This is arguably a cleaner approach that avoids worst case thinking. However, it makes a further assumption that we control the model, but this is reasonable in a lot of settings.

The choice of delta: clear typo of using n in definition? It is not written elsewhere that M’s weights are a single vector in R^p. The shifts are very limited, even for linear models, I would at least expect \delta_{min} and \delta_{max} to depend on each feature, is that extension easy to do?

There is a big assumption of approximating a non-linear model by a linear one locally, I am not sure how much weight I put on the validity of LIME explanations.

Algorithm 1 is an algorithm on the individual level, and so can be run independently which is nice.

In theorem 1, I don't understand what does it mean for x’ to lie on the original data manifold. In parentheses it is written that x’ is sampled from a normal distribution, the same as that of x, is that sampling independent? If so, then it is not possible that x’ is a recourse as x’ is only a function of x (which can be random, but the theorem says “THE” recourse which implies it is a deterministic map to add to the confusion). In the proof it is not clear how x’ is initially obtained, because if x’ is a recourse then it is a given that w^Tx’ >0! But you add that as a condition in \omega.

There is no analysis to the performance of the given algorithm, as section 4 assumes to be able to fully solve the optimization problem. I would like to see some argumentation for why the algorithm performs well theoretically.

In table 1, M1 and M2 validity results are very promising, but the lack of discussion on what does the increase in average costs imply is a bit unsettling. I would have liked to see in the experiments what is the lower limit on costs to achieve this high of M1&M2 validity, without context, I am unsure how bad the increase in avg cost is. Figure 2 helps tell me that the increase in avg cost is not too high because with a mean shift of 1, validity becomes very low, so that x’ was not on the extremes. I would have liked to see visual examples in the synth setting of original points x and recourses x’ for the different methods to understand what is happening.


----------------
Post response: I have read the authors response, the authors have given good responses to some of the concerns I had, my recommendation however remains a weak accept

**Time Spent Reviewing:**

5

---

> ### Author Response · Authors · 2021-08-12
> **Response to Reviewer efp6**
>
> We thank the reviewer for their insightful comments. Below, we address specific questions raised by the reviewer.
>
> **Our Contributions and Connections to Prior Work:** We are glad that the reviewer appreciates the quality of this work. To further underscore our contributions, we would like to clearly highlight the key differences between our work and Rawal et al. [1].
>
> At a high level, Rawal et al. [1] ask the question “Are state-of-the-art recourse algorithms robust to model updates which occur due to shifts in the data distribution?”. To address this question, they design an evaluation framework and carry out an extensive empirical analysis with state-of-the-art approaches to demonstrate that existing recourse algorithms are in fact not robust to model updates which occur due to shifts in the data distribution. They also provide some supporting theoretical results to demonstrate that counterfactuals/recourses closer to the decision boundary are more likely to be invalidated upon model updates.
>
> In light of this finding, our work focuses on addressing a key follow up question which is “How can we generate recourses that are robust to arbitrary model shifts?”.  We introduce this problem and propose the first known solution to address this problem. To this end, we introduce a novel framework called RObust Algorithmic Recourse) (ROAR), as part of which we formulate a novel optimization problem and develop algorithms to optimize our objective. In addition, we carry out a detailed theoretical analysis to not only establish a lower bound on the probability of invalidation of recourses generated by state-of-the-art algorithms but also bound the additional costs incurred by the robust recourses output by our framework. We also carry out extensive experimentation using the evaluation framework put forth by Rawal et. al. [1] to demonstrate the efficacy of our framework ROAR.
>
> As clearly highlighted above, the two works are distinct and their relationship follows the precedent set by other seminal explainability papers - e.g., “Sanity Checks for Saliency Maps” (NeurIPS 2018) [2] highlights critical challenges associated with saliency maps, and “A Simple Saliency Method That Passes the Sanity Checks” (ICLR  2020) [3] proposes a solution to address these challenges.
>
> **Theorem 1:**
> In theorem 1, we assume x’ is any instance from the data distribution that has label one (i.e., positive label) w.r.t. the original decision boundary. So, it is not independent of the decision boundary. We recognize that this may have been unclear and will update the draft to reflect this.
>
> **Algorithm guarantees:**
> We chose to analyze the ROAR objective and the corresponding optimal solution (Theorem 2) instead of our optimization procedure because we wanted to first answer the question of whether we can make theoretical claims about our conceptual formulation. Furthermore, our optimization procedure is inspired by adversarial training and employs gradient descent to solve a min-max problem. This is a popular and already well studied area in machine learning and optimization, and we didn’t consider providing convergence guarantees about these general purpose algorithms to be within the scope of this work. In addition, prior research has already demonstrated that using multiple restarts alongside gradient descent style approaches provides reasonable solutions to non-convex problems resulting in meaningful recourses [8,9]. Given all this, we chose to analyze the ROAR objective and the corresponding optimal solution instead of our optimization procedure.
>
> Analyzing Algorithm 1 carefully and improving the optimization framework is certainly important future work. While constrained algorithms with better convergence properties are available in case of convex (in x)/concave (in delta) loss functions, any such guarantees are challenging in non-convex/non-concave case.
>
> **Increase in Average Cost:** While ROAR based recourses are substantially more robust than other baselines, they can be slightly more costlier than recourses output by CFE [9] and AR [6]. While these baselines result in low cost recourses, they do so at the expense of generating unrealistic recourses that are off manifold [10].
>
> On the other hand, recourses generated by our ROAR framework consistently have lower costs than those output by another baseline MINT [10] which is a very popular state-of-the-art recourse baseline (Table 1, Section 5.2) that leverages causal graphs to generate realistic recourses. Given this, we do not consider the increase in average costs due to ROAR prohibitively high.
>
> **Practicality of recourse:**
> There has been a huge surge in the literature on algorithmic recourse recently [6,7,9,10]. This is mainly due to the fact that several regulatory agencies including EU General Data Protection Regulation (GDPR) have drafted guidelines which enforce the “right to explanation” for individuals adversely impacted by algorithmic decisions. Furthermore, while adverse action notices which notify customers who were denied credit are common in the US and the law (The Equal Opportunity Credit Act) requires that such notices be justified further, this does not always happen in practice and has often been criticized. The field of algorithmic recourse is often regarded as an effective solution to tackle the aforementioned real world problems [6, 9].
>
> Our framework which provides the first known solution to generate recourses that are robust to real world model shifts further paves the way for making algorithmic recourse more practical in real world applications. This is because in practice, models are regularly updated, and unless a recourse method takes this into account, like our proposed method ROAR, it will not generate recourses [6,9] that end users can rely on.
>
> **Extensions:** Thank you for the suggestions! While multi-class and probabilistic recourse are interesting ideas for future work, in this paper we consider the notions and conventions of recourse commonly adopted in the literature on algorithmic recourse [9].
>
> **Why not update the model constrained by the fact that old recourses must still hold?:** Thanks for making this point. We, in fact, considered two alternative solutions: a) Update the model as desired but all those individuals who were previously provided with recourse will still be guaranteed a favorable outcome. b) Update the model while including constraints to ensure that previously offered recourses are still valid (as pointed by the reviewer).
>
> Note that both of these scenarios would potentially incur huge monetary losses to the relevant stakeholders (e.g, banks). In case (a), banks may be required to guarantee credit to customers that are potentially not creditworthy under the new model and thereby risk losing money. In case (b), the trained model may be a suboptimal model and not reflective of the current data distribution thereby accruing larger errors under the shifted population. There are no incentives for stakeholders such as banks to adopt such practices which could potentially lead to huge monetary losses.
>
> **Choice of $\Delta$:** The definition of $\Delta$ is general and does not make assumptions on the model class. To clarify, we cannot be robust to every possible model shift and hence allow restricted perturbations as defined by the set  $\Delta$ and therefore assume a vector space of the parametrization of the ML model. Here, $n$ refers to the cardinality of the parameter space i.e. $n \triangleq |\mathcal{W}|$ - we will make this explicit in the definition. However, altering $\Delta$ such that $\delta_{min}$ and $\delta_{max}$ depend on each feature is an easy extension to ROAR - thank you for the suggestion!
>
> **Limited form of model shifts that apply only to linear models:** We would like to clarify the misconception that we only analyze a limited form of model shifts that only apply only to linear models. Any model shift (irrespective of why it occurs) can be characterized by a change in model parameters. Specifically, such shifts can be characterized as additive shifts of model parameters. While we instantiate our objective for additive parameter shifts to linear models, we successfully employ this approach for deep neural networks as well using local linear approximations from LIME (See NN row in Table 1/Figure 2). Note that this approach has been employed throughout the recourse literature [6,7].
>
> We thank the reviewer again for their thoughtful comments and feedback. We hope we addressed all your questions/concerns/comments adequately. In light of our clarifications, please consider increasing your score to accept. Please let us know if we can provide any further details and/or clarifications.
>
> *References:*
>
> [1] Rawal, Kaivalya, Ece Kamar, and Himabindu Lakkaraju. "Can I Still Trust You?: Understanding the Impact of Distribution Shifts on Algorithmic Recourses." arXiv:2012.11788 (2020).
>
> [2] Wang, Yuanhao, and Jian Li. "Improved Algorithms for Convex-Concave Minimax Optimization." NeurIPS 2020.
>
> [3] Daskalakis, Constantinos, Stratis Skoulakis, and Manolis Zampetakis. "The complexity of constrained min-max optimization." STOC 2021.
>
> [4] Wang, Yuanhao, and Jian Li. "Improved Algorithms for Convex-Concave Minimax Optimization." NeurIPS 2020.
>
> [5] Daskalakis, Constantinos, Stratis Skoulakis, and Manolis Zampetakis. "The complexity of constrained min-max optimization." STOC 2021.
>
> [6] Ustun et al., “Actionable Recourse in Linear Classification,” FAccT 2019
>
> [7] Rawal & Lakkaraju, “Beyond Individualized Recourse: Interpretable and  Interactive Summaries of Actionable Recourses,” NeurIPS 2020
>
> [8] Dick et al., “How many random restarts are enough”. Technical report, 2014.
>
> [9] Wachter et al., "Counterfactual explanations without opening the black box: Automated decisions and the GDPR." Harv. JL & Tech., 2017.
>
> [10] Karimi et. al., "Algorithmic recourse: from counterfactual explanations to interventions". FAccT 2021.

---

> > ### Author Response · Authors · 2021-08-20
> > **Empirical discussion of Theorem 2 (additional response for Reviewer efp6)**
> >
> > Thank you for your suggestion. In response to your comment, we have evaluated analytically the theoretical upper bound on the costs incurred by the recourses output by ROAR (Theorem 2). To this end, we leveraged the recourses output by CFE (Wachter et. al.) for a logistic regression model with L1 costs to compute the right hand side of our bound in Theorem 2. We compared these theoretical bounds with the empirical costs incurred by the recourses output by ROAR. Please find both the empirical costs as well as the theoretical upper bounds on the costs for different datasets in the table below. Note that the empirical costs in the table below are analogous to the Avg Costs reported in Table 1 in the main paper (we are reporting results on a subset of trials here). We will include these results in the final version of the paper.
> >
> > | Dataset | ROAR costs (Empirical value) | Theoretical Upper Bound ($\eta=0.1$)  |
> > |-------------|-----------------------|-----------------------|
> > | Correction Shift (German Credit)  |   3.056$\pm$0.272        |   4.620$\pm$0.174 |
> > | Temporal Shift (SBA case) |     4.807$\pm$0.146|   29.950$\pm$0.70 |
> > | Geospatial Shift (Student performance) |     14.174$\pm$0.333 |   32.137$\pm$0.524  |
> >
> > Note that in our response to Reviewer aXPK we have also included the empirical discussion and implications of Theorem 1. We hope that these additional results address your concerns regarding the empirical discussion of our theoretical results. In light of our rebuttal and these new responses, we hope the reviewer considers increasing their score to accept.

---

> > ### Comment · Reviewer_efp6 · 2021-08-31
> > **reply to response**
> >
> > Connections to Prior Work: thank your for the detailed comparison and outlining the differences.
> >
> > Increase in Average Cost: Thank you for the follow-up experiment, the upper bound is certainly helpful.
> >
> > Practicality of recourse: I am not exactly convinced that the "right to explanation" translates to a recourse rather than a why explanation, but nonetheless I am convinced that algorithmic recourses could become practically relevant in the future and that your work tries to pave the way for that. But I think it is not the effort that will best lead to recourses becoming practical, I think the bigger problems are that of interpreting and explaining recourses to the human individual.
> >
> > Choice of $\Delta$: I assume the results generalize naturally to this case by taking the bound with respect to the largest feature deviation.
> >
> > Limited form of model shifts that apply only to linear models: that makes total sense thank you!
> >
> > Why not update the model constrained by the fact that old recourses must still hold?: I would like this discussion to be included in a revised version of the paper.

---

> > > ### Author Response · Authors · 2021-09-01
> > > **Response to Follow Up Comments of Reviewer efp6**
> > >
> > > Thank you so much for the follow up. We are very glad to know that the reviewer found our prior responses helpful. Few other comments:
> > >
> > > **Practicality of recourse**: We agree with the reviewer’s assessment -- while the “right to explanation” may only relate indirectly to recourse as of now, algorithmic recourse could become practically relevant in the near future and works such as ours are attempts at paving the way for the same. Furthermore, seminal works such as [1] argued that algorithmic recourse has a strong ethical basis, and therefore has a very high potential to be employed in real world applications.
> > >
> > > While our work addresses one of the key challenges pertaining to the practicability of recourses, we agree with the reviewer that interpreting and explaining recourses to human individuals is also key to making algorithmic recourse practical. To this end, evaluating the interpretability of recourses output by our framework ROAR and other state-of-the-art approaches via user studies is an important direction for future work. We will include a discussion about this in the final version.
> > >
> > > **Choice of $\Delta$**: Yes, our results generalize naturally to this case by taking the bound with respect to the largest feature deviation. We will make a note of this in the final version.
> > >
> > > **Discussion on why not update the model**: We will include this discussion in the final version of the paper.
> > >
> > > Thank you so much for your time and effort in helping with the review of our paper. We hope that we have addressed all your comments/questions adequately. In light of these clarifications, we sincerely hope the reviewer considers increasing their score to accept.
> > >
> > > **References**:
> > >
> > > [1] Suresh Venkatasubramanian & Mark Alfano, “The philosophical basis of algorithmic recourse,” FAccT 2020.

---

### Official Review · Reviewer_aXPK · 2021-07-16

**Rating:** 6
**Confidence:** 4

**Summary:**

This paper proposes an algorithm for the automatic generation of counterfactual examples, that are robust to a certain amount of model shift, e.g., by retraining. The probability of invalidation of counterfactuals and the cost of robust counterfactuals are analyzed theoretically and the algorithm is evaluated on 3 tabular real-world and on synthetic data sets.

**Limitations And Societal Impact:**

The authors mention some limitations, e.g., higher counterfactual costs, but there could be discussion on additional limitations, e.g. that the argmax problem in Algorithm 1 may not be accurately solved in general, which may prevent convergence and on the computational cost that come with the proposed aproach or a general solution.

**Main Review:**

The formulation of the Counterfactual problem under predictive multiplicity is already discussed in the literature (see for example Pawelcyzk). Nevertheless, the proposed method for generating robust counterfactual examples under model shift is novel, so this work advances the state-of-the-art.

While the evaluation results show that the method leads to more robust counterfactuals, the evaluation was conducted on relatively small datasets (with 1000, 2102, 649 lines respectively). Benchmarking on a larger dataset (e.g., Give Me Some Credit, 150k lines) would additionally highlight the scalability of the method. The synthetic dataset evaluation supports the findings, but the setup is very simple. (Maybe cut the synthetic data part for more real data set experiments).

Unfortunately, the theory part has some issues. In general, the theorems only consider very special cases (linear classifier and simple exponential-family distributions) and implications for the actual methodology are not clear. Furthermore, the interpretation of the theoretic results is missing. The derived quantities are not explained or put into context. There are some theoretical problems that should be clarified by the authors (see below). In general, even if the derivation was correct, the unintuitive condition (no interpretation given) and the appearance of the beta value (which should be optimized out), makes it hard to see the implication of Theorem 1.

The bound in the second theorem linearly grows with the diameter of the dataset D, so the order of magnitude is the same as using a plain Lipschitz bound if c is Lipschitz-continuous. There are also some points in the derivation, which should be clarified.

Theorem 1:
1. $x'$ is assumed independent of the decision boundary; this is not realistic.
2. What is the meaning of $\beta$? Can it be maximized?

Theorem 2:
1. $\delta$ is the worst case for each counterfactual (this is a coupled min max problem, so it is not enough to solve each problem on its own!)  which can be very different for $x'$ and $x''$. If you assume the same $\delta$ for both cases, your argument is not valid.
2. In equation 33 onwards, it should be clearly marked, that the result is only valid with probability 1-$\eta$
3. The bounds that lead to equation 34 are not valid in general. Assume a Dirac distribution (or very narrow normal distribution, i.e. around zero). Then $E_\mu[log(1+exp(-z))] = log(2)$ but $E_\mu[z]=0$


The acronym "ROAR" is already taken and might be misleading (see https://research.google/pubs/pub47088/).

In conclusion, I find the paper is well written and addresses a relevant problem, but the theory part needs further consideration before it is ready for publication.

**Time Spent Reviewing:**

5

---

> ### Author Response · Authors · 2021-08-11
> **Response to Reviewer aXPK**
>
> We thank the reviewer for their insightful comments. Below, we address specific questions raised by the reviewer.
>
> **Comparison with Pawelcyzk et al. [1]:** We appreciate your observation that our work is novel with respect to existing literature. To further underscore our contributions, we would like to note that Pawelcyzk et al. [1] analyze the behavior of existing recourses under predictive multiplicity (i.e., the scenario that multiple different classifiers give almost equal solutions). On the other hand, our work deals with the problem of generating recourses that are robust to real world model shifts (where the shifted models would most likely give different solutions). So, the problem that we deal with is distinct from predictive multiplicity, a lot more challenging, and has implications for making algorithmic recourse practical in real world settings. Furthermore, we provide the first known solution to this critical and challenging problem (note that Pawelcyzk et al. [1] provide no solutions to the challenges associated with predictive multiplicity). We will clarify this distinction in our final version.
>
> **Datasets:**  Since the goal of this work is to generate recourses that are robust to real world model shifts, we evaluate our approach w.r.t. model shifts that commonly occur in the real world. More specifically, we consider model shifts that occur as a consequence of real world data distribution shifts. To this end, we would require datasets that capture different kinds of real world data shifts (e.g., temporal, geo-spatial etc.). This restricts the kinds of datasets that we can employ in our evaluation because not all real world datasets out there capture data shifts. While we are aware of other larger datasets such as the Give Me Credit dataset, we could not incorporate them in our evaluation because they do not capture data shifts. We will include this rationale in our final version.
>
> **Assumptions and Interpretation:** While we do make linearity and exponential family assumptions in our theoretical analysis, this is an accepted practice in the recourse literature. For example, the theoretical analysis in Ustun et al.’s [2] seminal paper makes linear model assumptions as well.
>
> Furthermore, we believe that these assumptions do not narrow down the scope of our results because recent literature in algorithmic recourse commonly considers local linear approximations of non-linear models. For instance, Ustun et al. [2] argue that their approach and the accompanying theory (which relies on the linear model assumption) can be readily applied to non-linear models by first generating local linear approximations of any given non-linear model using algorithms such as LIME (Ribeiro et. al.). We envision our results to be applicable to non-linear models as well in a similar manner.
> We also agree that our results show that Theorem 2 cost bound grows linearly with D, and will therefore be similar to making Lipschitzness assumptions over c. We will add this interpretation to our paper as you have suggested.
>
> Algorithmic recourse is still a nascent subfield of explainable AI with very little theoretical grounding. In addition, there are no other methods for generating recourses that are robust to model shifts, let alone theoretical analyses of them. Hence we believe that our contributions can form a critical building block for future work. We are more than happy to try and improve the quality of our bounds. Please let us know if you have any further suggestions regarding this.
>
> **Theorem 1:** We assume x’ is any instance that has label one (i.e., positive label) w.r.t. the original decision boundary, so it is not independent of the decision boundary. We recognize that this may have been unclear and have updated the draft to reflect conditioning on the event that $f(x’) > 0.5$ (or an appropriate decision threshold).
> For the gaussian case, we obtain a chernoff-type bound on the error function given by a function parametrized by $\beta$. We agree that it can be optimized to obtain a tighter bound; this optimal value is $\beta_opt = 1.080$ for lower bound. We will discuss this in the paper.
> Theorem 2: The $\delta$ in Theorem 2 is the worst-case for $x’’$ specifically. While it is true that the corresponding delta would be different for the non-robust recourse $x’$, our goal was to only focus on the added cost for $x’’$ by virtue of being robust for shift $\delta$ (specific to $x’’$). Therefore we do not need to consider the $\delta$ which $x’$ is robust to, and further, we only need to consider the difference in costs. Hence we compared the cost of using $x’$ when the shift corresponds to $\delta$(the maximum shift $x’’$ is robust to). You do raise a valid point that for this fixed $\delta$, the inequality requires an additional condition of convexity in $x’’$ so that Equation 24 is valid. We have added this to the draft.
>
> Equation 33 onwards are now marked as high probability statements as you suggested.
>
> Thank you for raising the point about bounding $E\mu[log(1+exp(−z))]$ where $z = (w+\delta)^Tx$’ and $x’$ is the recourse that is not robust to shifts $\delta$. We have made the following updates so that the bounds are valid even for the case you mention. To do so we have restricted the setting to the case where $x’$ is Gaussian so that  $(w+\delta)^Tx’$ is also a gaussian random variable. In particular, here are the updates:
>
> $\begin{align}
> E_{\mu}[\log{(1+\exp{-z})}] &= E_{\mu}[\frac{\ln{(1+\exp{-z})}}{\ln{2}}] \\
> &\leq \frac{1}{\ln{2}}E_{\mu}[\exp{-z}] \\
> \end{align}$
>
> Thus our bound holds so long as $E_{\mu}[\exp{-z}] < \infty$. For a Gaussian distributed $z \sim \mathcal{N}(\mu, \sigma^2)$, this can now be bounded by
> $\frac{\sqrt{2\pi\sigma^2}}{\ln{2}} \exp{(\mu - \frac{\sigma^2}{2})}$, which for a Normal random variable is $\frac{\sqrt{2\pi\sigma^2}}{\ln{2}}$ and should hold even for a narrow/dirac distribution. We have updated the draft to reflect this. Interpreting this, we now see that the bound is significantly loose if $\mu \ll 0$ and improves when $\mu \gg 0$. Note that here, the bound cannot be improved significantly by conditioning on the event that $f(x) > 0.5$ or equivalently that, $w^Tx’ \geq 0$ since we are concerned with bounding a quantity that is a function of the shifted decision boundary $w+\delta$. We have also added these interpretations as you suggested.
>
> **ROAR acronym:** Thanks for bringing this to our notice. We will update our acronym in the final version to avoid any confusion.
>
> We thank the reviewer again for their thoughtful comments and feedback. We hope we addressed all your questions/concerns/comments adequately. In light of our clarifications, please consider increasing your score to accept. Please let us know if we can provide any further details and/or clarifications.
>
> $\textit{References}$
>
> [1] Pawelczyk et al., “On Counterfactual Explanations under Predictive Multiplicity,” Proceedings of the 36th Conference on Uncertainty in Artificial Intelligence (UAI 2020)
>
> [2] Ustun et al., “Actionable Recourse in Linear Classification,” Proceedings of the Conference on Fairness, Accountability, and Transparency (FAccT 2019)

---

> > ### Comment · Reviewer_aXPK · 2021-08-16
> > **Thank you for the updates and the clarifications**
> >
> > Thank you for the updates and clarifications. Please also review the implications of Theorem 1 on the rest of the results in your paper. Assuming that you will do this and that you will also address the issues raised by the reviewers in the final version, I will increase my score.

---

> > > ### Author Response · Authors · 2021-08-19
> > > **Response 2 to Reviewer aXPK**
> > >
> > > Thank you so much for your positive response and for increasing your score. To understand implications of Theorem 1, we evaluated the bounds analytically and compared them with our empirical results. Please find the empirical results and theoretical lower bounds (Theorem 1) for invalidation probabilities of Wachter et. al.'s CFE method with the logistic regression model and L1 cost below.
> > >
> > > | Dataset | Empirical Value | Theoretical Lower Bound  |
> > > |-------------|-----------------------|-----------------------|
> > > | Correction Shift (German Credit)    |        0.46 $\pm$ 0.27        |       0.49|
> > > | Temporal Shift  (SBA case)   |           0.69 $\pm$ 0.09       |               0.50|
> > > | Geospatial Shift  (Student performance) |          0.71 $\pm$ 0.09        |       0.36|
> > >
> > > Note that the empirical invalidation probabilities are basically complements of M2 validity results from Table 1 in the main paper, and are computed as (1.0 - M2 Validity). It can be seen from the table above that our empirical results satisfy the theoretical lower bounds thus validating these bounds. In addition, it can also been seen that our bounds are quite tight for the dataset capturing correction shifts. We will include these results and address all the questions/concerns raised by all the reviewers in the final version of the paper.

---

### Official Review · Reviewer_Xovy · 2021-07-27

**Rating:** 7
**Confidence:** 4

**Summary:**

This paper deals with the robustness of algorithmic recourse options given to users by common methods in the field, as models get updated over time. In particular, unlike previous works, this one proposes a solution that aspires to withstand distributional shifts in training data over time, space, correction to measurements, etc. The proposed solution, ROAR, is inspired by adversarial training. The work also introduces a lower bound on probability of invalidation of recourse options of models trained normally, as well as upper bound the added cost the output of ROAR might incur. Finally, the authors compare the performance of ROAR, as well as to a causal version of ROAR (combined with Karimi et al.'s MINT) to those of existing popular methods, and offer a sensitivity analysis, demonstrating on a synthetic dataset the effect of distributional shifts on validity of recourse options.

**Ethical Concerns:**

None.

**Limitations And Societal Impact:**

The authors did not discuss limitation of their work, or much of its societal impact. Even the proposed future work citing "tradoff between recourse robustness and cost to better understand impact..." has been already somewhat done in this work, and its sister manuscript Rawal et al. However, given that most of the work deals and is motivated clearly by societal impact of algorithmic decision-making/decision-support systems, I believe its impact in this area is overall positive.

**Main Review:**

The paper seems overall novel to me, bearing great similarity to Rawal et al. (https://arxiv.org/abs/2012.11788v2), but being the more complete work of the two, promising greater contribution to the literature in my opinion. The proposed method and theoretical results seem like a good start to deal with the problem of algorithmic recourse and its validity over model updates.

Strengths:
1. Novel method proposed to deal with a recently identified possible concern in a relevant and impactful area of study.
2. Initial theoretical foundation proposed is interesting.

Weaknesses:
1. Choices in method and experimental setting could be discussed and justified more carefully (I'll include suggestions below).

Questions to authors:
1. (minor) in section 2, you quickly describe your connection to other works on adversarial robustness. It seems to me Wachter et al. 2017 were of the first to mention the connection between counterfactual explanations (precursor to algorithmic recourse) and adversarial examples. Might be worth mentioning that connection briefly.
2. (minor) line 126 - the objective function seems to directly relate to a classification setting, yet you write after it "denotes a differentiable loss function (e.g., mean squared error loss..."; why not just name binary cross entropy directly?
3. Section 3.2, choice of \Delta -- both here and in the following, the authors consider shifts in terms of *small* perturbations to parameter space or gradient of the model. What is *small* in this context? Can we bound it? what if not all shifts can be described in terms of additive perturbations of this kind? Can we demonstrate the additivity in parameters space or gradients following the shifts demonstrated in real world data in the experiment section?
4. line 165-6 - "in our framework, the perturbations are applied to model parameters as opposed to data samples". It is not immediately clear to me why one should choose to apply the perturbations in this case on the models rather than data samples. While I can see why it's reasonable, could the authors explain their choice more? I may have missed the point.
5. line 179-80 - "...when the model shifts can be characterized as additive shifts to model parameters". Similar to the question above -- can all perturbations occurring as a consequence of distribution shifts be characterized as additive shifts?
6. Section 5.1, Real world data - from the description of the SBA dataset and the student performance dataset it wasn't clear to me whether the distributional shifts as proposed would be quite enough to be called "temporal" and "geospatial" shifts, respectively. what is the temporal shift exactly for SBA? It doesn't seem like there were 2 different periods of collection, but a single continuous one? Not entirely clear to me what would be the data for M1 vs M2 without further discussion. As for the student performance dataset, are two collection points into schools in the same country reflect enough variation? Would be interesting to see what both these examples mean in terms of actual shifts occurring in model parameters/gradient, as well as in terms of variables, covariance matrices, etc. to get a better sense of what such shifts mean in practice.
7. Discussion of Karimi et al. and causal methods in the experimental section - first, the authors only compare results to MINT and MINT-ROAR for the German credit score, citing the fact that only this dataset "has an existing DAG" via Karimi et al. However, this DAG of the German credit dataset is also not official or appeared with the original dataset. It was proposed by Karimi et al. as a conjecture, which the authors of this work could have done with the other remaining dataset in this section if they wanted to achieve a more comprehensive comparison. On a related but different point, in line 344-45, the authors state "MINT and ROAR-MINT seem to exhibit highest avg costs.... likely because adhering to the causal graph incurs additional cost"; this is almost a tautological sentence, just repeating the experimental result, without attempting to provide any explanation of why that might happen. I believe one possible hypothesis could be that recommending interventions on nodes in the root or close to root nodes in a causal graph might incur additional costs to downstream features, as the effect needs to be propagated through the DAG. Non-causal methods would assume independence of features, thus no downstream propagation of intervention effects, and only "local" cost to intervened nodes would appear, although this is very likely unrealistic.


**Time Spent Reviewing:**

7

---

> ### Author Response · Authors · 2021-08-12
> **Response to Reviewer Xovy**
>
> We thank the reviewer for their insightful comments. Below, we address specific questions raised by the reviewer.
>
> **Our Contributions and Connections to Prior Work** We are glad that the reviewer appreciates the novelty and the significance of this work. To further underscore our contributions, we would like to clearly highlight the key differences between our work and Rawal et al. [1].
>
> At a high level, Rawal et al. [1] ask the question “Are state-of-the-art recourse algorithms robust to model updates which occur due to shifts in the data distribution?”. To address this question, they design an evaluation framework and carry out an extensive empirical analysis with state-of-the-art approaches to demonstrate that existing recourse algorithms are in fact not robust to model updates which occur due to shifts in the data distribution. They also provide some supporting theoretical results to demonstrate that counterfactuals/recourses closer to the decision boundary are more likely to be invalidated upon model updates.
>
> In light of this finding, our work focuses on addressing a key follow up question which is “How can we generate recourses that are robust to arbitrary model shifts?”.  We introduce this problem and propose the first known solution to address this problem. To this end, we introduce a novel framework called RObust Algorithmic Recourse) (ROAR), as part of which we formulate a novel optimization problem and develop algorithms to optimize our objective. In addition, we carry out a detailed theoretical analysis to not only establish a lower bound on the probability of invalidation of recourses generated by state-of-the-art algorithms but also bound the additional costs incurred by the robust recourses output by our framework. We also carry out extensive experimentation using the evaluation framework put forth by Rawal et. al. [1] to demonstrate the efficacy of our framework ROAR.
>
> Algorithmic recourse is still a nascent subfield of explainable AI with very little theoretical grounding. In addition, there are no other methods for generating recourses that are robust to model shifts, let alone theoretical analyses of them. Hence we believe that our contributions can form a critical building block for future work.
>
> As clearly highlighted above, the two works are distinct and their relationship follows the precedent set by other seminal explainability papers - e.g., “Sanity Checks for Saliency Maps” (NeurIPS 2018) [2] highlights critical challenges associated with saliency maps, and “A Simple Saliency Method That Passes the Sanity Checks” (ICLR  2020) [3] proposes a solution to address these challenges.
>
> **Other specific questions:**
>
> (1-2) **Minor:** These are excellent points and we will make the recommended changes in the final paper.
>
> (3) **Choice of $\Delta$:** The magnitude (or “smallness”) of the shifts that we consider is determined by the parameter $\delta_max$. As discussed in Section 5.1, we set this parameter to 0.1 in our real world experiments. We also carried out sensitivity analysis to understand the effect of varying $\delta_max$ (See Section D.2 in Appendix) and observed that as $\delta_max$ increases, ROAR remains robust but cost increases. Note that the idea of considering small perturbations has precedent in the adversarial robustness literature where small perturbations to data samples are leveraged for training predictive models that are robust to dataset shifts.
>
> (4) **Perturbations are applied to model parameters:**
> Applying perturbations to model parameters is one of the central ideas of our approach.
> To understand why we apply perturbations to model parameters as opposed to data samples, let us first consider the literature on adversarial robustness [5]. The high level goal of this literature is to learn predictive models that are robust to changes in the data. To this end, perturbations are applied to data instances (to capture unknown changes to the data that may occur in the future) and the resulting perturbed instances are leveraged for learning predictive models that are robust to changes in the data.
>
> On the other hand, the primary goal of our work is to generate recourses (or counterfactuals) that are robust to changes in the models. So, we address this problem by applying perturbations to model parameters (to capture unknown changes to the model that may occur in the future) and the resulting perturbed models are leveraged for learning recourses that are robust to changes in the models. Furthermore, in our work, we consider arbitrary model shifts which can occur due to any reason (and not just due to data distribution shifts). So, we refrain from capturing/modeling any kind of data shifts and focus exclusively on capturing model shifts. This is why we apply perturbations to model parameters (as opposed to data samples).
>
> (5) **Additive shifts to model parameters:** Irrespective of the cause, a model shift is characterized by a change in model parameters. We would like to note that arbitrary linear model shifts can be captured via additive perturbations in the parameter (coefficient) space, and we leverage this property in our work. To understand why this is true, let $W_1$ be the coefficients of the original linear model and $W_2$ be the coefficients of the shifted linear model. Let $\delta = W_1 - W_2$. There will always exist some $\Delta$ such that $\delta \in \Delta$ can be applied as an additive perturbation to $W_1$ so that desired shift to $W_2$ is captured.
> In addition, it is not too far-fetched to think about modeling shifts in parameter/gradient space corresponding to complex non-linear models using additive perturbations [4]. Furthermore, our empirical results (Section 5.2 in particular) clearly demonstrate that we are able to effectively capture model shifts induced by complex real world data shifts by modeling them as additive perturbations. This confirms that additivity can effectively capture model shifts in real world settings.
>
> (6) **Section 5.1, Real world data:** Due to space constraints, we describe how we generate M1 and M2 for temporal and geospatial shifts in Section D.1 in the Appendix under “Setting and Implementation Details”.
>
> In our experimental setup for training M1 and M2, we attempt to mimic a typical real world setting where we first have some data using which we build a model (M1), we then collect more data (e.g., over time) which we leverage in conjunction with previously available data to build a better model (M2). In line with this set up, for the SBA dataset (temporal shift), we use data from  1986-2006 to train M1 and data from 1986-2012 to train M2. For the Student Performance dataset (geospatial shift), we train M1 on data from one school district, and then train M2 after incorporating data from another school district as well.
>
> To determine what the actual model shifts look like, we compute the L2 distance between M1 and M2 coefficients (in case of logistic regression). The average distance (averaged across 5 trials as part of 5 fold cross validation) along with the standard deviation for each dataset is as follows:
> German Credit (correction shift) - $0.28 \pm 0.08$
> SBA (temporal shift) - $1.22 \pm 0.09$
> Student Performance (geospatial shift) - $2.20 \pm 0.26$
>
> We observe that for larger L2 distances (i.e., larger model shifts), the difference in robustness between ROAR and baselines grows, with ROAR based recourses being substantially more robust. For a detailed analysis on the impact of the degree of distribution shift on the robustness of the recourses generated by our framework, please refer to our experiments in Section 5.3.
>
> (7) **Causal methods:**  While the causal graph for the German credit dataset put forth by Karimi et al. [6] only contained 4 features, the other real world datasets we leverage in this work contain 20+ features. This made intuiting an accurate causal graph far more complex for these datasets, and using an inaccurate causal graph would render MINT ineffective. So, we refrained from making up our own causal graphs for other real world datasets. Regarding the high cost of MINT and ROAR-MINT, we agree with your hypothesis and will reword that sentence. Our original intention was to convey the following: Since non-causal recourse methods do not have to adhere to the underlying causal structure when finding counterfactuals, they can generate relatively lower cost counterfactuals even if those counterfactuals may not correspond to realistic data instances. This is one of the key reasons why we observe lower average costs in case of non-causal recourse methods compared to causal recourse methods. We will include this discussion in the final version.
>
> We thank the reviewer again for their thoughtful comments and feedback. We hope we addressed all your questions/concerns/comments adequately. In light of our clarifications, please consider increasing your score to accept. Please let us know if we can provide any further details and/or clarifications.
>
> *References*
>
> [1] Rawal et al., “Can I Still Trust You?: Understanding the Impact of Distribution Shifts on Algorithmic Recourses,” https://arxiv.org/abs/2012.11788v2.
>
> [2] Adebayo et al., “Sanity Checks for Saliency Maps,” 32nd Conference on Neural Information Processing Systems (NeurIPS 2018).
>
> [3] Gupta et al., “A Simple Saliency Method That Passes the Sanity Checks,” ICLR 2020.
>
> [4] Ross et al., “Improving the Adversarial Robustness and Interpretability of Deep Neural Networks by Regularizing Their Input Gradients,” 32nd AAAI Conference on Artificial Intelligence (AAAI-18).
>
> [5] Madry et al., “Towards deep learning models resistant to adversarial attacks,” ICLR 2018.
>
> [6] Karimi et al., “Algorithmic Recourse: from Counterfactual Explanations to Interventions,” Proceedings of the 2021 ACM Conference on Fairness, Accountability, and Transparency (FAccT 2021).

---

> > ### Comment · Reviewer_Xovy · 2021-08-31
> > **Thank you for your detailed reponse**
> >
> > I would like to thank the authors for their great responses to my questions, as well as the other reviewers. To me, the only remaining points of consideration relate to the following questions, many based on efp6's comments:
> >
> > 1. "why not update the model constrained by the fact that old recourses must still hold?": your response was thoughtful, but pointed out that both this approach and the one chosen in the paper could incur cost to the modeling body. Therefore, I'm not sure you sufficiently justified your choice? Perhaps I missed a subtlety?
> >
> > 2. Increase in recourse cost: I interpret your response to efp6 to say that you are doing better than MINT, and not terribly worse compared to the other two baselines (CFE and AR), and therefore there is no cause for concern. However, I think there is a good explanation to why your approach seems to be doing better than MINT, which may not be an entirely fair comparison (see Q7 in my original review). Your rebuttal regarding this point doesn't seem entirely satisfying.
> >
> > 3. "expect \delta_{min} and \delta_{max} to depend on each feature": this is a very interesting point. I hope you will include this extension in a future version.
> >
> > 4. "validity of LIME explanations": another great point, that I think your most compelling response to was essentially that this is common practice in algorithmic recourse literature. Please discuss this possible limitation in your next version, for example in light of recent work addressing LIME's validity, e.g. [Zhao et al. UAI 2021, BayLIME](https://www.auai.org/uai2021/pdf/uai2021.342.pdf).
> >
> > However, all these points are somewhat minor. Another main point of discussion is the relation of this work to Rawal et al. Given that Rawal et al. was not officially published in a peer-reviewed manner, I tend to think this work should be accepted and in essence be combined with Rawal et al. (as the latter is essentially just pointing to the phenomenon -- that I think is to be expected given the nature of algorithmic recourse methods -- which this work actually grapples with). Therefore, assuming you will address the points above, I will increase my score from 6 to a 7.

---

> > > ### Author Response · Authors · 2021-09-02
> > > **Thank You and Response to Follow Up Comments of Reviewer Xovy**
> > >
> > > Thank you so much for the positive feedback to our response and for increasing your score to 7 (accept). Please find below our responses to your remaining questions:
> > >
> > > **Why not update the model constrained by the fact that old recourses must still hold?**:
> > >
> > > We would like to clarify a subtlety here. Let us consider the scenario where a bank leverages an ML model to determine which customers should be given loans (i.e., which customers are creditworthy). Let us assume that recourses are provided to those customers whose loan applications are rejected by the ML model.
> > >
> > > In the real world, there are likely to be changes in the underlying data distribution which would in turn require the model to be updated (so that the model is still accurate on the new shifted data distribution). If we update the model constrained by the fact that old recourses must still hold, we are essentially saying that the bank is required to guarantee credit to customers that are potentially not creditworthy according to the new (shifted) data distribution. This could imply huge monetary losses for the bank since customers who are not creditworthy according to the new data distribution may not be able to pay back their loans. Therefore, this strategy is not really viable for the bank from a practical standpoint.
> > >
> > > Our approach, on the other hand, does not constrain the model in any way but instead focuses on generating recourses that are likely to be valid w.r.t. various model shifts. This ensures that customers who implement our recourses are likely to remain creditworthy even when the underlying data distribution (and consequently the model) changes. This alleviates or at least minimizes the risk of monetary losses for the bank.
> > >
> > > Given the above rationale, banks (and other stakeholders) are more likely to adopt our framework ROAR in practice over the alternate approach of updating the model by constraining old recourses to hold.
> > >
> > > **Increase in recourse cost**:
> > >
> > > We agree with the reviewer's intuition about why our approach (which is a non-causal recourse method) incurs lower costs than MINT (which is a causal recourse method). By comparing our costs with that of MINT, we are merely suggesting that if the costs associated with MINT (which are higher than our recourse costs) are considered acceptable in real world settings, there is no reason to believe that our recourse costs are prohibitively high. We should have clarified this more clearly. Apologies for any confusion.
> > >
> > > **Expect \delta_{min} and \delta_{max} to depend on each feature**:
> > >
> > > Yes, this is a simple extension to our framework. We will definitely include this extension and results with it in the final version of the paper.
> > >
> > > **Validity of LIME explanations**:
> > >
> > > We will definitely discuss the possible limitations with LIME in light of the recent work by Zhao et. al. (UAI 2021). We will also experiment with the BayLIME approach and will discuss these new results in the final version.
> > >
> > > We will discuss all the aforementioned points in the final version and address connections with Rawal et. al. as per your suggestions. Thank you so much for your time and effort in helping with the review of our paper. We really appreciate the increase in score.

---

### Decision · Program_Chairs · 2021-09-27

**Decision:**

Accept (Poster)

**Comment:**

The expert reviewers for the most part appreciated the paper and were largely positive. There were many important issues raised but the reviewers felt the authors responded appropriately and suggested a reasonable path to addressing these within the scope of this review process. The authors are expected to carefully address the points raised by reviewers in a final version, including as they outlined in their response. This includes the points raised by about the theory and assumptions as well as justifying choices in the model -- the answers from the reviewers were adequate and they need to be incorporated into the paper.